# Access and Benefit Sharing and the Sustainable Trade of Biodiversity in Myanmar: The Case of Thanakha

Alessandra Giuliani [1,*], José Tomás Undurraga [2], Theresa Dunkel [3] and Saw Min Aung [4]

1 School of Agricultural, Forest and Food Sciences BFH/HAFL, Bern University of Applied Sciences, 3052 Zollikofen, Switzerland
2 Faculty of Environment and Natural Resources, University of Freiburg, 79085 Freiburg im Breisgau, Germany; jose.undurraga.q@gmail.com
3 AGROSCOPE, Plant-Production Systems, 1964 Conthey, Switzerland; theresa.dunkel@outlook.fr
4 HELVETAS Swiss Intercooperation, Yangon 1062, Myanmar; SawMin.Aung@helvetas.org
* Correspondence: alessandra.giuliani@bfh.ch

**Abstract:** The global demand for biological resources to use as natural ingredients in diverse products is rising rapidly. This creates investment opportunities for nature-based products, creating pressure on and threats to biodiversity and its associated traditional knowledge (TK). Myanmar's great biodiversity is attractive for scientific investigations searching for natural substances for pharmaceutical, cosmetic, and other uses. Myanmar is amid profound political and economic changes, exposing the country to risks and opportunities. The recent opening to world trade put its rich biodiversity and TK under severe threat. One of the local natural biodiversity products is Thanakha, which is traditionally used for skincare. This study investigates the current and planned regulations and practices managing Access and Benefit Sharing (ABS) in Myanmar, focusing on one of the potential BioTrade products: Thanakha. A qualitative and quantitative survey was conducted through in-depth interviews with 37 key informants and 35 Thanakha farmers. The results show that while the current research and development activities at the Thanakha manufacturing level could trigger ABS obligations, the low awareness about ABS requirements and the lack of traceability raise uncertainties for its potential implementation. The implementation of BioTrade principles and ethical sourcing to promote the sustainable trade of Thanakha, as well as the implementation of ABS, would lead to the protection of biodiversity and TK, and the improvement of local livelihoods.

**Keywords:** ABS Institutional framework; BioTrade; Burmese thanaka; traditional knowledge; genetic resources; natural ingredients; value chain; Nagoya Protocol; *Hesperethusa crenulate/Limonia acidissima*

## 1. Introduction

### 1.1. Access and Benefit Sharing, and the Use of Natural Ingredients

The world's demand for biological resources as natural ingredients is growing rapidly. This creates new opportunities for investment and the production of nature-based products, but also places pressure on natural resources, thus creating threats for local biodiversity and its associated traditional knowledge [1,2]. On the other hand, because of the increasingly globalized economy, international trade associated with specific value chains of products demanded in developed countries has become one of the main drivers of species loss in biodiversity-rich nations, in particular in developing countries [3,4].

The agricultural, pharmaceutical, chemical and cosmetic industries benefit directly from biodiversity and its services. Genetic compounds of plant and animal origins can be of high value when they are used in pharmaceutical products. The exploration of potentially valuable genetic resources (GR) and natural biochemical compounds is referred to as bioprospecting [5]. Many cosmetic and personal care products are based on natural products like essential oils, pigments or surfactants, or are based on wild species derivatives such as saponins, flavonoids, amino acids, antioxidants and vitamins [6]. While genes,

species and ecosystems are crucial for this industry, serving as raw materials and regulators of natural processes, the current biodiversity loss rate scenario threatens the industry through an increase of operational costs because of higher costs for the sourcing of raw material, together with more restrictions on access to biological resources [7]. Hence, there is a need to promote sustainable trade systems that reduce the threats to livelihood development and biodiversity protection in rural territories [8].

In 1992, the international community took a big step towards the protection of biodiversity and adopted the Convention on Biological Diversity (CBD), with three objectives: (i) the conservation of biodiversity, (ii) the sustainable use of biodiversity, and (iii) the fair and equitable sharing of the benefits derived from the use of genetic resources (GR) [9,10]. Equitable sharing is considered a precondition to the attainment of biodiversity conservation and sustainable use [5]. In 2010, the CBD party members signed the Nagoya Protocol as a tool to implement the third objective [11]. A key concept under the protocol is Access and Benefit Sharing (ABS), which is related to the ways in which users access to GR and its associated traditional knowledge, and how the benefits that arise from their use are shared between users and providers [10,12]. The providers are the states with sovereign rights over the natural resources under their jurisdiction, and the Indigenous people and local communities (IPLCs) in cases when traditional knowledge (TK) associated with GR is involved. The protocol's goal is to protect the IPLCs' interests by recognizing the importance of their TK linked to the use of GR [12]. The users can be researchers, universities or industries (i.e., food, pharmaceutical, cosmetic, agriculture) which access the GR for research or commercial purposes. The goal of an effective ABS system is to ensure that users have suitable access to GR and its associated traditional knowledge, sharing in return the benefits derived from their use with the country of origin and the traditional knowledge owners. In relation to the implementation of ABS, the CBD recognizes two main principles regulating the access to GR by the users, and how the benefits arising from their use are fairly and equitable shared with the providers. National governments are entitled to implement the following conditions to ensure a fair and equitable ABS framework [10]:

-   Access to GR and TK is granted through a "prior informed consent (PIC)" given by the national authority of a provider country to a user.
-   The sharing of the benefits derived from the use of GR and associated TK must be agreed upon according to some conditions between the providers and users, through the so-called "mutually agreed terms" (MAT) [5].

The implementation of the basic measures of the Nagoya Protocol in the participating countries will release widespread monetary and non-monetary benefits for the providers of GR. Some of these benefits should be reinvested in the conservation and sustainable use of the biological resources from which the genetic resources were obtained. This will fulfil the three objectives of the CBD [5,12].

Investment and trade activities play a fundamental role in helping to maintain biodiversity and promote the sustainable development of rural livelihoods [5]. In 1993, the United Nations Conference on Trade and Development (UNCTAD) raised awareness about the link between global trade and sustainable development. UNCTAD BioTrade Initiative was launched in 1996 to promote trade and investment in biological resources, and was aligned with the CBD's objectives for further sustainable development, poverty alleviation and biodiversity conservation [13]. BioTrade included "activities related to the collection or production, transformation, and commercialization of goods and services derived from native biodiversity (genetic resources, species and ecosystems) according to criteria of environmental, social and economic sustainability" [14], as defined by a set of principles; some of them were strictly related to CBD provisions on ABS. The UNCTAD BioTrade Initiative (BTI) seeks partnership along three strategic lines: enabling a policy framework for BioTrade, value chain enhancement, and market creation for biodiversity products and services [14]. Since 2003, the BTI has hosted the BioTrade Facilitation Program (BTFP), which focuses on enhancing sustainable bio-resource management, product development, value adding processing and marketing [15]. BioTrade initiatives aim to create

business models for sustainably sourced, traceable and value-added natural ingredients to conserve biodiversity through the sustainable trade of natural ingredients, in order to increase the livelihood benefits of the rural population of the countries where the resources originate [16].

### 1.2. Biodiversity in Myanmar

Myanmar is rich in biodiversity. The majority of the country is positioned inside the Indo-Myanmar Biodiversity Hotspot, which is ranked within the top 10 hotspots globally for irreplaceability and in the top five for threats, thus supporting high levels of biodiversity and endemism [17,18]. The Country hosts wide Key Biodiversity Areas (KBAs, Figure 1), i.e., sites contributing significantly to the global persistence of biodiversity [19].

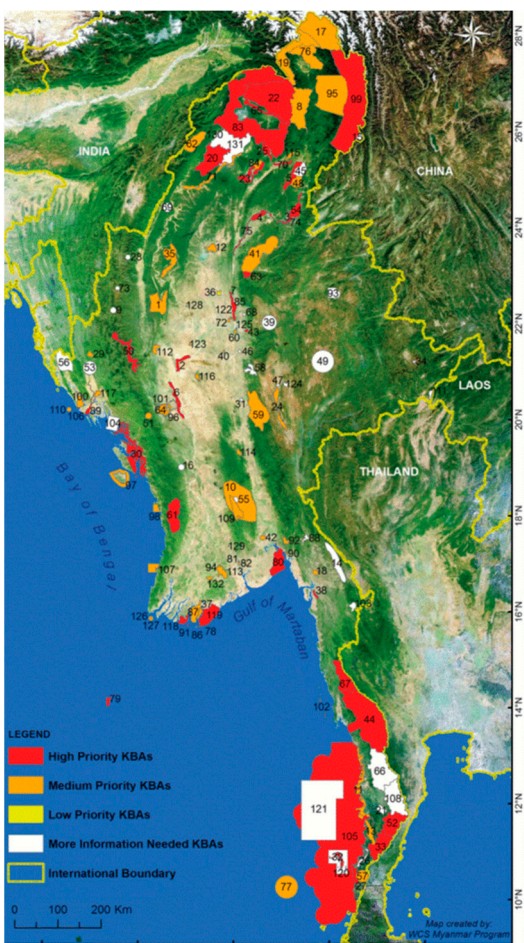

**Figure 1.** Key Biodiversity Areas (KBAs) in Myanmar (Wildlife Conservation Society Myanmar).

Because of its large extension, diverse types of topography and habitats can be found here, from the alpine mountainous region in the north to the dry deciduous forest in the central area, to the mangrove and tropical rainforest in the south [18]; it thus contains a wide variety of species and ecosystems—in other words, high biodiversity levels—providing a home to around 300 mammal species, more than 1000 birds, 370 reptiles and 18,000 plants (of which 39 species of mammals, 45 birds, 21 reptiles and 38 plants are globally threatened). The plant species that have been recorded in Myanmar include more than 800 orchid species, 80 bamboo species, numerous rattan species and more than 800 medicinal plant species; the latter indicate the potential for biodiscovery through new research. The country is rich in wild varieties of rice, and oilseed crops, legumes, medicinal orchids, ginger, chili, onion and potato, which have also been given conservation priorities by government research institutions in Myanmar due to their high value for utilization [20].

Myanmar relies heavily on ecosystem services and biodiversity for the livelihood of its population and economic growth. The agricultural sector—which represents 36% of the Gross Domestic Product (GDP) and 13.3% of the country's total export earnings, and employs 61.2% of the total labour force—depends highly on ecosystem services provided by natural ecosystems, such as pollination, the maintenance of soil structure and fertility, nutrient cycling and hydrological services, and biological pest control [4]. Forests are also fundamental to the socio-economic wellbeing of the people of Myanmar, and for the timber trade. Currently, over 70% of the country's population are rural, and are reliant on forest resources for food, shelter and warmth [21].

Myanmar's high levels of biodiversity make it an attractive location for scientific investigations to find natural substances for medicinal, cosmetic, pharmaceutical and other uses. Myanmar is in the midst of profound political and economic changes that are exposing it to both risks and opportunities. The recent opening to world trade (in 2011) put the country's rich biodiversity resources and its traditional knowledge at a risk: Myanmar's globally important biodiversity hotspot is severely threatened [22]. The largest encompassing threats are the overexploitation and continuing loss of forest, and the loss of the TK of local communities associated with genetic resources. Deforestation in Myanmar is driven by hydropower development, infrastructure construction, shifting cultivation, the extraction of wood for fuel, and aquaculture development. It is reported that Myanmar had a net loss of 310,000 hectares annually (2000–2010), an annual rate of 0.93%, one of the ten highest annual net losses of forest area in the world [23,24].

Overexploitation at a subsistence level is also caused by the fact that the TK of local communities associated with genetic resources is being lost rapidly due to the changes in traditional lifestyles and the replacement of the TK associated with the medicinal usage of biological resources by modern medicines. Nevertheless, Myanmar is still relatively intact in terms of its forest areas in comparison to the region [23,24].

Although Myanmar is a country with high levels of biodiversity, where one main concern is the conservation of species and natural ecosystems [12], unfortunately, Myanmar's biodiversity is being threatened in an alarming way [4,25]. High rates of biodiversity loss can be observed all over the country. Myanmar is the third country in terms of threatened species within the Indo-Burma Biodiversity Hotspot, owning 16% of all of the threatened species in the region [26]. The major threats to biodiversity in Myanmar are habitat clearance and degradation, the illegal trade of endangered species, the unsustainable use of natural resources, and vulnerability to Climate Change and related changes to ecosystems. Some of the underlying causes are poverty, increasing consumption patterns due to the recent economic growth, and the global demand for natural resources [25]. Though the conservation of nature and wildlife is regarded as one of the national priorities—incorporated within which are biological resources—with the problems Myanmar is facing in peace building, good governance, and reconciliation, biodiversity and natural resources remain lower in the agenda priorities. On the other hand, natural resource management, and among this biodiversity, is one of the main reasons for conflict: minor ethnic groups live near and have access to more rare biodiversity resources in biodiversity hot spots [4,27].

For the last 2000 years, Myanmar's people have been using the bark of *Hesperethusa crenulata* (syn. *Limonia acidissima*), a common tropical plant found in the Indian subcontinent and Southeast Asia [28], to produce Thanakha, a nature-based product traditionally used as a paste for skin care, and as a cosmetic product [28,29].

A traditional practice among most Myanmar women is to rub a piece of Thanakha wood with a little water on a plate of stone that is specially made—called a "kyauk-pyin"—to produce a watery paste. The Thanakha paste is applied on the whole face, making cosmetic, attractive patterns which are distinctive of Myanmar's culture and tradition: commonly circles on the cheeks and forehead. It was, and still is, used for protection against tropical sunlight and air-transmitted harmful substances in the belief that Thanakha can enhance skincare benefits [30]. Its other dermatological purposes include acne treatment and prevention, skin cooling and whitening, scar fading, repelling mosquitos, and odor

deterrence [31]. Thanakha was identified as a potential high-export BioTrade product for Myanmar [16]. While the local conditions are good to develop Thanakha for export markets, the product is not well known or widely used outside of Myanmar and a few neighbouring countries. The widespread use of Thanakha in high-value markets interested in BioTrade will likely require the R&D of new products based on this native resource and traditional knowledge associated to its efficacy and uses. The R&D activities and process could trigger the ABS requirements under the Nagoya Protocol [32]. Although Myanmar signed the Nagoya Protocol in 2014, its implementation is still at a very early stage. According to Access and Benefit Sharing (ABS) Clearing House [33], a website hosted by the Convention on Biological Diversity (CBD), Myanmar set up an ABS National Focal Point (NFP) and a Competent National Authority (CNA) that published an Interim National Report on the Implementation of the Nagoya Protocol (NR) in July 2018.

The study aims to investigate the current and/or planned regulations, systems and practices that will regulate and manage ABS for the sustainable trade of biodiversity in Myanmar. In addition, an investigation with a particular focus on one of the potential BioTrade products in Myanmar (Thanakha) was conducted. The study is not an ABS Gap analysis.

The study includes two specific objectives: (i) to analyse the current state and potential context of Access and Benefit Sharing in Myanmar, and (ii) to analyse the potential for the sustainable trade of biodiversity in Myanmar's ABS context.

## 2. Materials and Methods

A mixed research approach was used based on qualitative and quantitative methods for the collection of secondary and primary data collected between 2018 and 2019.

In order to analyse the first specific objective, the study started with a comprehensive review of the secondary data, specifically literature and information relevant to the subject of ABS in the context of Myanmar. The review helped us to get to know the existing information regarding the study topics in the area, and helped us to understand the institutional framework and the nature of the stakeholders involved. The literature review was based on the analysis of both books and peer review papers about the natural resources and biodiversity context in Myanmar and Access and Benefit Sharing related to use of natural ingredients, as well grey literature, i.e., project reports from international and local research, non-governmental and governmental organizations; other specific ad-hoc documents provided by scientists and the stakeholders interviewed during field work; and databases and information repositories like the Land, Agribusiness and Forestry Forum of Myanmar document repository (www.mylaff.org, accessed on 3 October 2021), which offers policy documents and a vast amount of published and unpublished grey literature based on study reports by international organisations and official governmental documents on the situation of ABS in Myanmar.

The first specific objective was also tackled with qualitative primary data collected through in-depth key-informant interviews, which were conducted with selected stakeholders using a questionnaire with semi-open and open questions. The interviews contributed to gain in-depth knowledge of Myanmar's implementation of ABS measures from the view of different key stakeholders, such as from the government, research institutes and NGOs, as well as key stakeholders involved in the Thanakha Value Chain. The key informants were identified in consultation with Helvetas BioTrade Team Myanmar and, from the literature review, the previous network and other experts in the field of ABS. In-depth interviews were carried out with 37 respondents from different stakeholder groups, as shown in Figure 2. The rate of response was 62.5%.

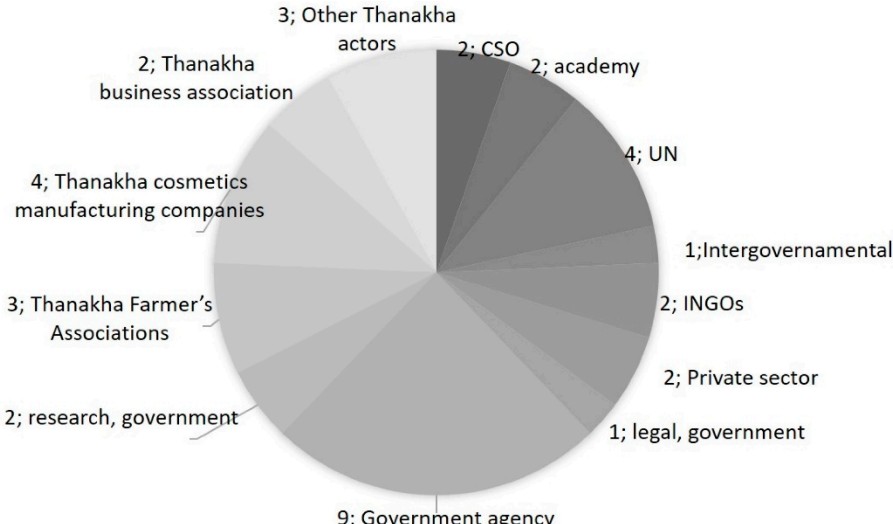

**Figure 2.** Number of actors per stakeholder group, from in-depth qualitative interviews.

In order to analyse the potential for the sustainable trade of biodiversity in the context of ABS in Myanmar, the study performed an in-depth analysis of the Thanakha Case Study. In the case study, quantitative and qualitative data were collected through structured interviews based on a questionnaire implemented at the Thanakha farmer level. This process contributed to the gaining specific knowledge related to Thanakha production and commercialization in the context of Myanmar's ABS implementation. The information collected through this instrument was also used to complement a relevant Thanakha Stakeholder and Value Chain analysis [32]. The survey included sections on (i) Household information, (ii) socioeconomic information, (iii) productive information, (iv) genetic resources and traditional knowledge, and (v) equitable benefit sharing. A total of 35 interviews were performed with respondents belonging to 5 townships. Because of the specificities associated with the different production areas, the sampling of the survey was focused on the inclusion of farmers from all of the main Thanakha production areas in Central Myanmar's Dry Zone (see Figure 3).

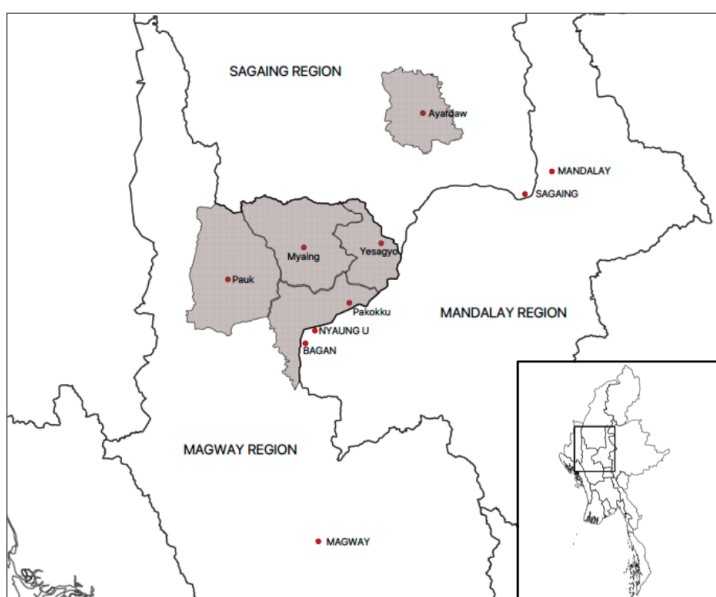

**Figure 3.** Study site for the quantitative data collection from Thanakha farmers.

The townships where the survey was implemented were selected as part of the BioTrade project by Helvetas Myanmar. In addition to belonging to Thanakha's main production area, the selection criteria of the townships focused on municipalities that already had farmers' associations in place and were collaborating with the project beforehand [32]. Given the time limitation, a purposive sampling was applied to include preferably market-oriented farmers, with different landholding sizes, located in convenient areas/linked to the project. A further small-scale survey was carried out with a purposive sample of 20 potential consumers in Switzerland and France, investigating Thanakha products' market potential. It aimed to identify the preliminary consumer interest and its potential commercialization in Europe on a case study basis. A case study is not a marketing study and it is not representative, but it gives a preliminary indication of the possible consumer attitude towards a new product.

A content analysis [34] was used for the analysis of the data collected through qualitative interviews with the key informants. In order to systematically structure larger texts from open questions, the software MAXQDA for Qualitative Data Analysis was used [35]. For the Thanakha market system, the main focus was to determine the current level of implementation of ABS or other measures, in order to analyze how well prepared the Thanakha Market System key players are to meet the requirements of equitable benefit sharing, such as those ones mentioned in the Ethical BioTrade Standard [32,36].

## 3. Results and Discussion

### 3.1. The Situation of Access and Benefit Sharing in Myanmar

This section reports the results related to the first specific study objective, i.e., to analyse the current state and potential context of Access and Benefit Sharing in Myanmar. The results are based on the literature review and primary data based on a quantitative content analysis of in-depth interviews with key informants from different stakeholder groups related to the ABS framework in Myanmar.

3.1.1. Institutional Framework for Biodiversity and ABS Matters in Myanmar

Myanmar has been party to the Convention on Biological Diversity (CBD) since 1995, and to the Nagoya Protocol since 2014 [37], being one of the first ASEAN countries. However, the first time Myanmar was involved in an ABS conference was in 2011. Myanmar joined the International Treaty on Plant Genetic Resources for Food and Agriculture (ITPGRFA) in 2002. The Ministry of Natural Resources and Environmental Conservation (MoNREC) is the focal point ministry for environmental and biodiversity-related matters, and it hosts the current national institutional framework dealing with these matters [20]. Currently, MoNREC Forest Department (FD) and the Environmental Conservation Department (ECD) are the focal departments on matters related to the CBD. ECD hosts the National Focal Point for the Nagoya Protocol, which is responsible for coordinating national-level efforts towards the implementation.

In 2011, Myanmar launched the National Biodiversity Strategy and Action Plan (NBSAP), with a pledge to genetic resource management, particularly with respect to crops and livestock [20]. Other key ministries collaborate with the work of the NBSAP.

A few high-level cross-departmental groups, such as the National Environmental Conservation and Climate Change Committee (NECCCC) and the National Biodiversity Conservation Committee (NBCC), participate in the environmental conservation policy dialogue [25]. Figure 4 depicts the current institutional framework dealing with matters related to the environment and biodiversity in Myanmar.

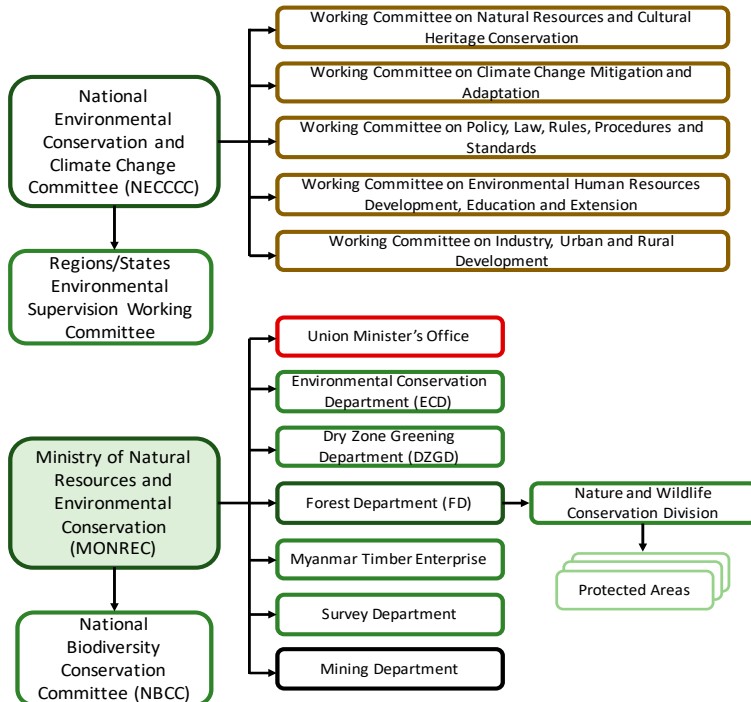

**Figure 4.** Environmental and biodiversity protection institutional framework in Myanmar (adapted from MCRB 2018 [25]).

### 3.1.2. Regulatory Framework and Current Developments Related to ABS

Although Myanmar became party to the Nagoya Protocol in 2014, a specific regulatory framework associated to ABS requirements is not yet in place. There is no law and policy directly and explicitly referring to access to GR and the fair and equitable sharing of benefits arising from their utilization or its associated traditional knowledge. Laws and policies to protect the environment—with some content related to ABS—are to be found in environmental, agricultural, livestock, forest, and traditional medicine marine resources and intellectual property rights (IPR) laws [25] (Appendix A Box A1). New expected policies aiming to reinforce the protection of biodiversity and ecosystem services are the new National Environment Policy (NEP), the Environmental Strategic Framework, and the National Climate Change Policy [25].

Another relevant sector for ABS in Myanmar is traditional medicine. There are about 7000 registered traditional medicine practitioners regulated by the Traditional Medicine Council Law (2000), and there are more than 12,000 Myanmar traditional medicinal drugs registered (Myanmar Food and Drug Board of Authority (FDBA), regulated by the Traditional Drug Law (1996)).

### 3.1.3. Current Changes Happening in the Legislation Related to ABS

The existing copyright law is 100 years old, originating long before the international agreements (World Intellectual Property Organization, WIPO; Trade-Related Aspects of Intellectual Property Rights, TRIPS). However, four IP laws are currently in the making [38]: the Trademark Law, the Patent Law, the Industrial Design Law and the Copyright Law. Once these new IP laws come into force, they will substantially change the current process of IP registration and protection in Myanmar. They will establish a new regime of an IP system, enhance the business environment, and provide better protection for IPR in Myanmar. For many years, the WIPO has been supporting and fostering the drafting of the trademark law. As in many countries, the WIPO is becoming progressively significant for reinforced IPR systems [5].

The Traditional Medicine Department of the Ministry of Health has a draft measure on the IP protection of traditional medicine which is currently pending in Parliament. This

would establish a law on the *sui generis* protection of traditional medicine. The Geographic Indication (GI) concept is also currently being discussed in some areas. Myanmar has no GI-certified products yet. Sometimes, the concept of GI is still not fully understood by different parties. Several studies report a variety of challenges related to the implementation of GIs in developing countries [39]. An attempt to develop GI products has recently been made, but the products have not been selected according to the GI principles (products linked to the terroir and tradition). An example was the focus on an introduced rice variety in an area where the variety was not known or used before. GI application to Thanakha products from Myanmar is under consideration.

### 3.1.4. Recent and Current ABS Supporting Activities in Myanmar

Collaborative efforts focussing on the implementation of the Nagoya Protocol and the development of an ABS system in Myanmar are currently occurring at different levels. Although some of the most relevant interventions are based on international cooperation initiatives, other types of collaborations are happening at the government and private levels [32]. Since 2011, international interventions have supported the implementation of ABS in Myanmar. One of the main objectives of the projects was to raise the level of awareness about ABS among governmental parties and the involved stakeholders through capacity building initiatives. These international initiatives also fostered most of the collaboration platforms which were created. Table 1 lists the projects supporting the national ABS implementation, including the draft of the national policy and institutional framework on ABS.

**Table 1.** Recent ABS-related projects in Myanmar.

| Project | Period |
|---|---|
| 1. ASEAN ABS Project<br>Building capacity for implementing CBD provisions on access to genetic resources and sharing benefits | 2011–2014 |
| 2.UNDP-China Trust Fund Project<br>Support for Ratification and Implementation of the Nagoya Protocol on Access and Benefit Sharing in ASEAN Countries | 2015–2016 |
| 3.UNDP/GEF Project<br>Strengthening human resources, legal frameworks and institutional capacities to implement the Nagoya Protocol | 2016–2019 |
| 4.UN Environment/GEF Project<br>Effective implementation of Myanmar's commitments under the Nagoya Protocol through enhanced policies on access and benefit sharing from utilization of genetic resources and associated traditional knowledge | Started in 2019 |

It is very interesting to mention that these international interventions, in addition to the planned outcomes, contributed to fostering further small-scale collaborations supporting ABS implementation [32]. Awareness-raising activities are going on, i.e., on biopiracy and NP-ABS principles among researchers and universities, and pilot projects for the implementation of NP-ABS in BRD, enhancing and powering some articles in MoU for further international collaborations.

### 3.1.5. ABS Institutional and Policy Framework

Another important objective of the recent and current international interventions mentioned in the above session is the support to the draft of the national policy and the institutional framework on ABS. At the end of the UNEP-China Trust Fund Project conducted by the ASEAN Centre for Biodiversity and through the UNDP-GEF V intervention, an ABS institutional and policy framework was identified in 2016, with the deliberation on the national focal point for ABS and the National Competent Authorities (NCA). The identified and approved structure of the ABS institutional and policy framework is the

following: MoNREC—the Environmental Conservation Department, ECD—was appointed as the ABS National Focal Point. The following institutes have been designated as NCA (Table 2), with the task of being responsible for granting access, or, as applicable, issuing written evidence that access requirements have been met. They will also be responsible for advising on the applicable procedures and requirements for obtaining prior informed consent (PIC) and entering mutually agreed terms (MAT) for the equitable benefits sharing.

**Table 2.** Proposed Institutional Framework drafted by the ASEAN Centre for Biodiversity (adapted from ACB 2016 [40]).

| Level | Entity | Description |
|---|---|---|
| NCA | • MoNREC Environmental Conservation Department (ECD) and the Forest Department (DF)<br>• Ministry of Agriculture, Livestock, Irrigation (MoALI) Department of Agricultural Research (DAR)<br>• Ministry of Health (MoH), Department of Traditional Medicine (DTM), and Food and Drug Authority<br>• MoALI Fisheries Department<br>• Ministry of Education (MOE), Department of Sciences and Technology | Designated by the National Focal Point |
| Coordination mechanism | Cabinet committee on ABS | Composed by the National Focal Point and the designated NCA to discuss and resolve national ABS issues |
| | Parliamentary Liaison | Cabinet Committee on ABS shall establish a liaison unit that will work with the Parliament on ABS issues |
| | Involvement with communities and other holders | Coordination mechanisms established to involve communities and other stakeholders, including the business sector and the academic community on ABS related issues |
| Implementing actions | Technical working group | Lead by the MoNREC ECD for identifying actions and resources necessary to establish and enhance the national institutional and policy ABS framework |

However, the framework is still in the making, and some nominees from various ministries for the ABS National competent authority are being confirmed. The ASEAN Centre for Biodiversity (ACB) endorsed that Myanmar ABS institutional framework should involve all of the relevant stakeholders, including governmental entities, the private sector and associations, and civil society organizations representing indigenous communities [40]. A number of permit systems to access GR in Myanmar exist from different governmental agencies (Appendix A Box A2).

3.1.6. Current Situation on Traditional Knowledge (TK) in Myanmar

Myanmar lacks a unique definition of TK. To cite an example, a product like rice wine, an alcoholic drink used on different special occasions (weddings, funerals, or to welcome people), which is an intrinsic part of traditional life from the Kayah valleys to the Chin hills [41], has an associated TK. The identification of the TK related to it would be very difficult, as manufactures in Chin, Kachin and Kahin State prepare it in a different ways (adding cereals to rice), but they claim the ownership of the TK. So, the question: How should we define TK for individual communities and specific products? In the field of traditional medicine (TM), some traditional medicine practitioners can provide instructions

on how to use the TM in different ways according to the indigenous uses of the place they are from. So, the question: Where does the TK belong to?

The existing law on IP mainly deals with trademarks, patents and copyright [42]. However, TK could be protected by copyright, as explained by the key informant from the Agricultural Research Department at the MoALI. In fact, people from different parts of the world argue whether genetic resources should be common heritage or should belong to a community [43].

In Myanmar, TK is normally transferred orally from one generation to another without being documented. There is no formal mechanism in place at the national level to document TK. State divisions have been mandated to preserve the TK related to medicine, though the effort is not considered to be fully successful. The Department of Traditional Medicine (DTM) of the Ministry of Health and Sport owns a database of traditional medicine which is only partially maintained, due to lack of human resources. The DTM is collecting an ancient record of TM formulations written on palm leaves, which are brought by some of the old TM practitioners to their library. Again, due to the flexibility in the formulation of TM according to different indigenous knowledge, and the way the knowledge is transferred from one generation to another, it is difficult to create a patent related to the TM. Gender-related aspects of TK exist, and they have not been studied yet.

According to ACB 2016 [40], the federal setup of Myanmar may offer an opportunity to provide a clearer and more definite policy framework that deals with the issues of indigenous peoples and local communities.

### 3.1.7. Possible Options Identified by the Interviewed Stakeholders

Many of the interviewees could not foresee any possible scenario for the development of the ABS system in Myanmar. Only a few key informants could describe the possible options, which are summarized as follows:

Creating a new specific ABS Law: MoNREC will prepare a draft ABS law and submit to the Union Government for its approval, and then to the bill will go to Parliament. At the same time, a law for the registration of TK is needed by MoNREC, supported by the new Ministry of Ethnic Nationalities and the Ministry of Border Affairs and the Border Area.

Amending existing laws—Modifying selected laws to incorporate ABS principles: This process would imply the amendment of selected laws to incorporate ABS principles, especially the Environmental Conservation Law, the Wildlife Law and the Seed Law. For the amendment, the participation of different stakeholders (including indigenous and local communities, IPLCs, and the academic and private sectors) is sought.

Developing an ad-hoc soft Law on ABS: According to the key informant suggestion from MoNREC ECD, it would be more advisable to develop a soft law first, and then enforcing it later. The interviewees reckon that reinforcing the legal framework is more important now, and that standard operation procedure can be enforced only once peace and stability is granted in the country. For a full enforcement of the laws there is a need to build the local communities' and ethnic minorities' trust in the Union Government. Local people and ethnic minorities' rights on GR and TK need to be protected. Customary rights must also be considered. Besides this, the development of a soft law on ABS will take about two years, so it would be much shorter than the process to develop a new specific law. Going through a soft law helps communities to increase the awareness about ABS before an ABS law is developed. The consultation conducted for the development of a soft law should help achieve the willingness and participation of rural communities and ethnic minorities.

The final decision about pursuing one of the above-mentioned three approaches, and on how to proceed, is up to the Union Government of Myanmar. In order to support the Union Government, more work is required to shed light on the needs, i.e., conducting a full ABS gap analysis which can build on the work already performed by the previous projects and activities, and the recommendations on whether an administrative, policy or

legislative measure is to be pursued. Based on this, Myanmar can decide which measure to pursue.

Among the difficulties foreseen for this process are (i) the lack of interest in ABS among the policy makers; (ii) the difficulty in organizing a consultation with different stakeholder groups, i.e., policy makers, local communities, ethnic groups, business, etc.; (iii) the remaining lack of trust among communities and the government though the efforts put in place. "Only awareness, peace and stability will lead to law enforcement: mutual trust among communities and government must be built first".

### 3.1.8. Overview of the Obstacles and Limitations Identified

A number of obstacles and limitations in relation to the ABS system's development in Myanmar have been identified through this study.

The present laws do not fully reflect the Nagoya Protocol requirements. Several new legislations are in the process of being approved, but they do not adequately incorporate the requirements of the Nagoya Protocol. Besides this, there is a lack of institutional communication among different authorities and stakeholders. Smith (2018) [44] reports that there is evidence from a number of countries that accessing the National Focal Points or Competent National Authorities in order to receive appropriate information is frequently problematic. Collaborations among institutions exist, but they are scattered and mainly based on personal relationships. The capacities related to the implementation of the national ABS system, such as the processing ABS access applications, negotiating ABS agreements, facilitating access to genetic resources, implementing compliance mechanisms and monitoring activities, etc., of the concerned entities are still scarce.

Due to the high turnover of staff in the Governmental institutions, a loss of institutional memory and the related capacity occurs. Financial limitations have also been mentioned as a constraint to develop the ABS system in the country.

Another problem identified is related to the past and current political situation, which fostered a general lack of mutual trust between communities and local/national authorities.

The lack of practical experience in establishing national and international partnerships, guided by ABS principles, limits the capacity of the negotiation of partnership agreements with foreign institutions. There is no previous experience of the formulation of partnerships between the communities and institutions that are involved in investigations and bioprospecting efforts that are guided by local operational procedures on ABS.

### 3.1.9. Limited Awareness of the Value of GR and Understanding about an ABS System

The level of awareness and understanding about CBD in general and the ABS system is still very low in Myanmar, even among governmental officials and academics. Even individuals from institutes which are concerned about ABS-related issues are reluctant to get involved in the discussion, as they think they are not knowledgeable enough or that ABS does not encompass their work. Some of the interviewees specified that there is lack of understanding of what ABS is at the level of decision makers and among Parliament members.

Similarly, there is limited information which has been made accessible related to existing and new opportunities for the utilization of genetic resources and bioprospecting projects that can attract research partnerships and additional resources. Assessing the potential value of GR in general is a problematic issue because of the still-limited knowledge about world biological diversity [5]. Without such information being available and accessible to research institutions and business companies, there will be limited interest and motivation for more systematic exploration and screening of genetic resources in Myanmar. A national, centralized and online knowledge management system to store information related to biological and genetic resources and associated TK is sought.

### 3.2. Thanakha and Access and Benefit Sharing in Myanmar

This sub-section illustrates the findings related to the second specific study objective, i.e., to analyse the potential for the sustainable trade of biodiversity in Myanmar's ABS

context, focusing on the Thanakha case. Both qualitative and quantitative data analysis are presented, following the literature, the survey with the Thanakha farmers and the small-scale study with key informants related to the potential export.

3.2.1. Thanakha in Myanmar

Thanakha is a semi-domesticated tropical plant (from the species *Hesperethusa crenulata* and *Limonia acidissima*) which is common in India and Southeast Asia. Because Thanakha could easily adapt to Myanmar's Dry Zone conditions, the tree has been traditionally grown in the central area of Myanmar, between Shwebo district in the Sagaing region and Pyay district in the Bagó region. For the last 2000 years, Myanmar people have used the bark of *Hesperethusa crenulata* or *Limonia acidissima* for the production of 'Thanakha paste', a nature-based product in the form of a white-yellow paste which is traditionally used by women (Figure 5) and children as a skin care and cosmetic product [29]. The traditional method of producing Thanakha paste is by grinding the bark of the tree stem against a slab stone, as shown in Figure 6 [45].

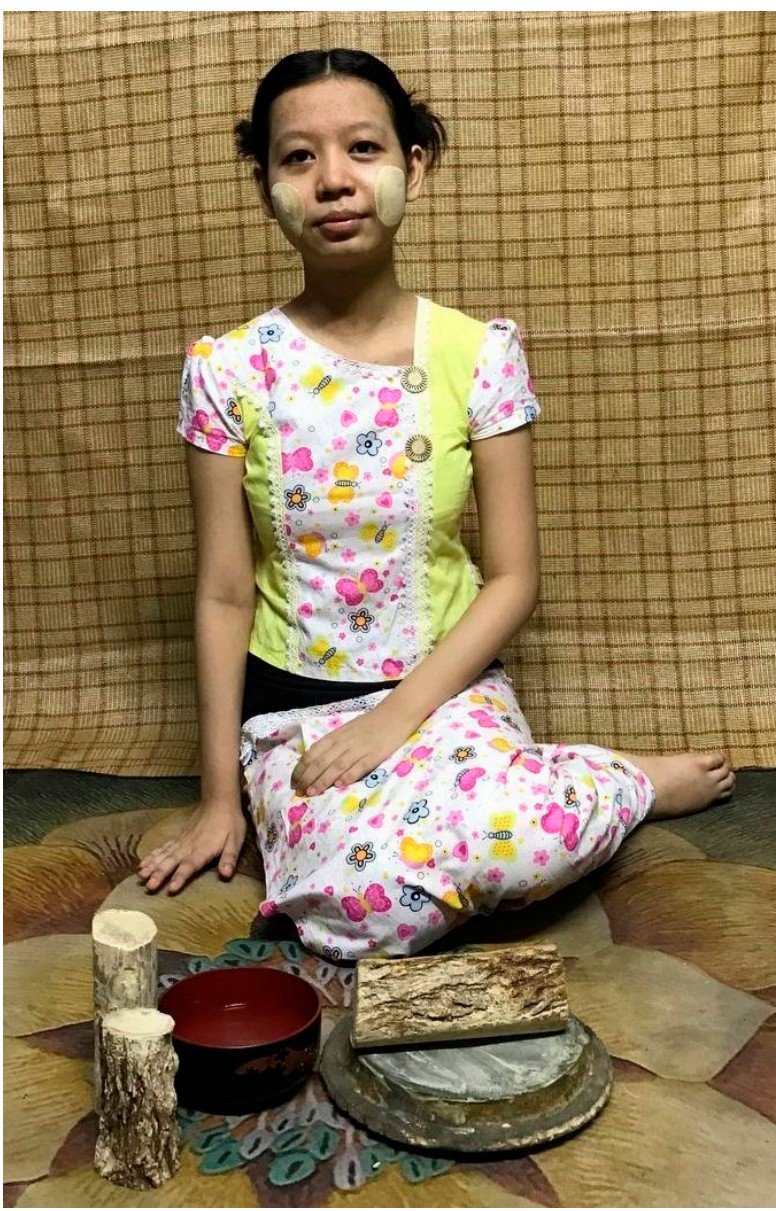

**Figure 5.** Myanmar woman wearing Thanakha (photo by A. Giuliani).

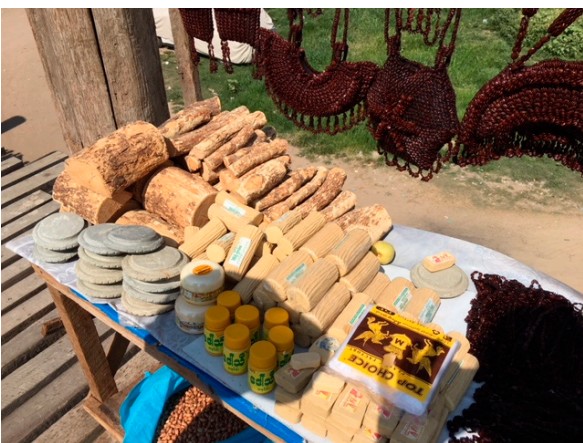

**Figure 6.** Traditional Thanakha bark and slab stones (photo by A. Giuliani).

Farmers from Myanmar's Dry Zone, particularly from Sagaing region, have been facing different challenges for several decades [46]. On the one side, the degraded soil quality together with unstable climatic conditions have been affecting production yields, and on the other side, market price fluctuations have impacted the commercialization process, creating uncertainties for further investment. In this context, Dry Zone farmers were pushed to adapt to this new environment of water scarcity and market fluctuations by introducing new cash crops that could secure their livelihoods through inter-cropping systems. Thanakha started to be commercially cultivated by farmers with the aim of diversifying their farming practices towards systems that could provide higher economic viability [45].

### 3.2.2. Thanakha Value Chain in a Nutshell and Its Involved Actors

According to the key informants and traditional literature, Thanakha has being produced, traded and used in Myanmar since ancient times [28]. Because of Thanakha being so deeply rooted in Myanmar's culture and people's daily routine, plenty of actors are involved in the production, processing and trading activities, creating a complex system of interactions at different levels of trade and value-added processes. While most of the production and trading activities are developed at the local level, processing, manufacturing and retailing activities are commonly performed either at the regional or national level. In terms of added value activities, two main processes were identified: the processing of raw Thanakha, mainly for production of Thanakha paste, and the manufacturing of more elaborate Thanakha cosmetics. Figure 7 illustrates the main leading market players currently involved at different core stages along the Thanakha value chain.

From a Market System Development perspective, several value chain actors have been identified and clustered by stakeholder groups, such as the direct value chain actors, the actors involved in the enabling environment, and the business service actors.

a. *Value Chain Actors*

- Farmers: Thanakha farmers are at the core of the Thanakha value chain for the production of the raw material, managing the local resources and owning the Myanmar thanakha-related traditions and culture.
- Intermediaries: Thanakha brokers and traders operate at different levels. At the local level, intermediaries source directly from the farmers, transport the goods, and sell to local processing plants and the regional wholesaler in the Pakokku township. At the national level, traders distribute raw Thanakha to the retailers and processors across the whole country. An important number of local traders are also farmers, producing and selling directly to consumers at local markets. The sales are mainly performed in traditional formats of different cut-stem styles. An important regional Thanakha market is located in Myaing



town, where customers from the township can buy locally produced Thanakha. Another important Thanakha market is located in Ayardaw town. The sale prices can vary depending mostly on the quality, but for a cut of stem of medium quality the price rounds out to 1500 MMK (USD 1). While the sourcing practices of these actors are mostly directly from farmers, the direction of the trader´s sales varies between local processors, regional wholesalers, and national processors. This group creates the most extended net of actors across the country, connecting farmers with processors, other intermediaries, manufacturers, retailers and finally customers. Unlike other Thanakha actors, the intermediaries are mainly women (Figure 8).

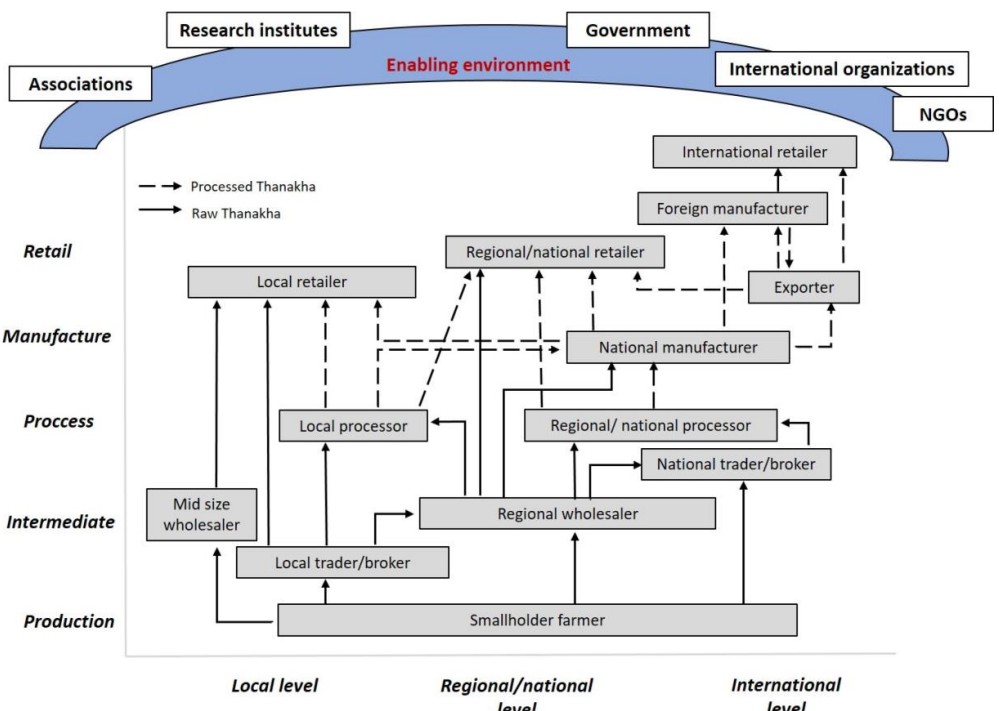

**Figure 7.** Thanakha value chain in Myanmar, with the value chain actors and enabling environment.

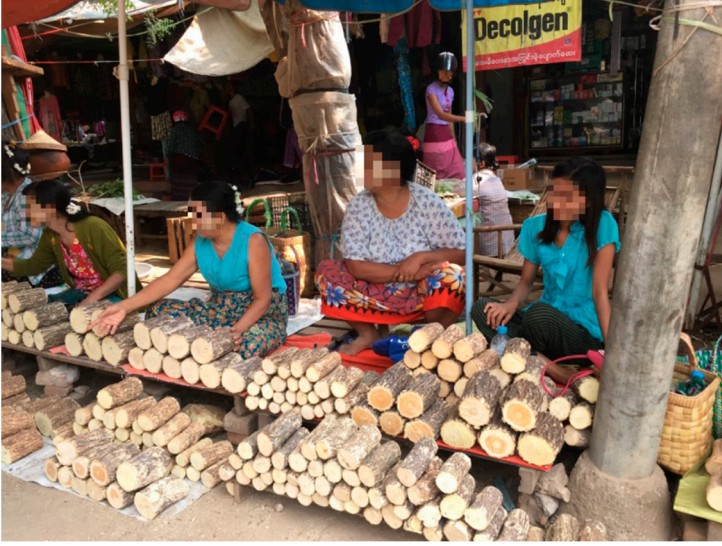

**Figure 8.** Women commercializing in Myaing Thanakha market (photo by Undurraga, 2018 [32]).

A single wholesaler from Pakokku Township is currently working with 50 intermediaries who buy from farmers and sell to him the raw Thanakha. This unique wholesaler delivers at a national level according to market demand. The main markets currently sourcing Thanakha from the Pakokku wholesaler are Mandalay, Yangon, Bago, Mawlamyine, and Sittwe. Besides raw Thanakha stems, this wholesaler also commercializes Thanakha branches to processors. The sale price is 200 MMK per *viss* of cut Thanakha cut branches, and the sale format is in sacs of 40 *viss*.

- Thanakha processors: Thanakha processors are commonly family-based small- to medium-sized companies located at the local or regional level. The business is based on transforming raw Thanakha into paste bars by grinding Thanakha branches using a wheel shaped "Taung Oo" stone that spins, driven by an engine-powered machine. Another business associated with local processing plants consists of peeling the bark from the stems to supply the manufacturing industry. The whole Thanakha paste production process consists on seven steps; first is the grinding of the Thanakha branches in the machine using water to obtaining, as a result, a muddy solution of water and Thanakha. Then comes the filtering process, leaving the solution on a filter-based structure for around 6 h; the complete draining of the water is done using a press system for approximately one hour, followed by a manual molding process once the paste is obtained, to finally pass through a sun- drying stage before the packaging of the final Thanakha paste bars. While local processors source most of their raw material from local traders, regional processors source mostly from regional wholesalers that distribute to almost the whole country. Both local and regional processors sell their final products directly to local and national retailers, although partly for demand-based cases, sales are also made directly to consumer.
- Thanakha manufacturers: Among the most important Thanakha value chain actors for value adding and the market value share are the manufacturers of Thanakha-based modern cosmetics. Mainly located in Yangon, these family-based businesses are currently an important driving force for the Thanakha industry, and are also involved in the expansion of the sector to other countries through export activities. Two major manufacturing companies currently operating in Myanmar are Shwe Pyi Nann, the largest Thanakha company in Myanmar, owning 125 acres of Thanakha plantation in Bagan, and Shwe Bo Minthamee, which has been involved in the Thanakha business sector since 1971. Although the business focus is still on manufacturing traditional Thanakha paste, they are also putting in efforts to develop Thanakha-based cosmetics and a variety of other skin care products. Both companies are currently exporting their products to Thailand, India, Malaysia, Indonesia, Philippines, Vietnam, Australia, Germany, and the USA. These companies do not source directly from farmers, but from a regional wholesaler from Pakokku Township and either local or regional processors supplying Thanakha bark.

b. *Enabling environment*

- Thanakha farmers' and traders' associations: Thanakha farmers' and traders' associations are an important element within the Thanakha Market System in Myanmar, as they represent the Thanakha farmers from the main productive areas at the township level. The associations from the Ayardaw (300 members) and Yesagyo (508 members) Townships were created in 2014, followed by the Myaing Township association (380 members) in 2015. The Pakokku Township association is in the process of being formalised. The associations' role is to support the development of new markets, to increase the Thanakha surface area by attracting more farmers to Thanakha production activities, and to support Thanakha farmers financially and with technical and marketing skills.
- Thanakha business associations: Two main associations in the Thanakha business sector exist. Both associations pursue the further development of the Thanakha industry in Myanmar. Its main role is to promote market expansion and encourage the export of Thanakha-based modern cosmetics. The Thanakha National

Federation involves farmers and traders through their respective associations, and three large manufacturing companies. The main role of the Thanakha National Federation is to facilitate a dialogue platform for the Thanakha value chain actors to discuss constraints from production to marketing.

- Government ministries and agencies: While the role of the state in the sustainable development of the Thanakha industry is crucial, currently, the promotion of the Thanakha sector is not one of Myanmar's government priorities. In any case, there are some initiatives driven by specific ministries promoting the development of Myanmar's Thanakha industry. Thanakha is categorized as a NTFP by the Forest Department, which issues permits for the transport of these resources, but farmers still claim to have problems in transporting unprocessed Thanakha. The Ministry of Commerce has facilitated the creation of the Thanakha Farmers' and Traders' Associations at the township level, and is supporting Thanakha-business-related companies with foreign R&D institutions.
- International organizations: Even if they are not closely related to the Thanakha Value Chain in Myanmar, there still is a group of international organizations supporting the sustainable development of biodiversity-based products in the region. UNCTAD is supporting Thanakha VC through the Regional BioTrade Project. GEF, UNDP, UNEP and ACB have been supporting the implementation of the Nagoya Protocol in the region, as well as the protection of traditional national products like Thanakha.
- Research institutes: Yezin Agricultural University and Yadanabon University in Mandalay are currently performing research on Thanakha, specifically on Thanakha paste's properties and its shelf-life, and about the traditional cultural aspects of Thanakha. Further studies are needed on the various properties of Thanakha in order to reveal its skin therapeutic potential [31].

All stakeholders play an important role and have an influence on the value chain of Thanakha. This study highlights the particular importance of the farmers' associations, as well as the main expectations of the interviewed farmers regarding the expected role of the associations (Figure 9).

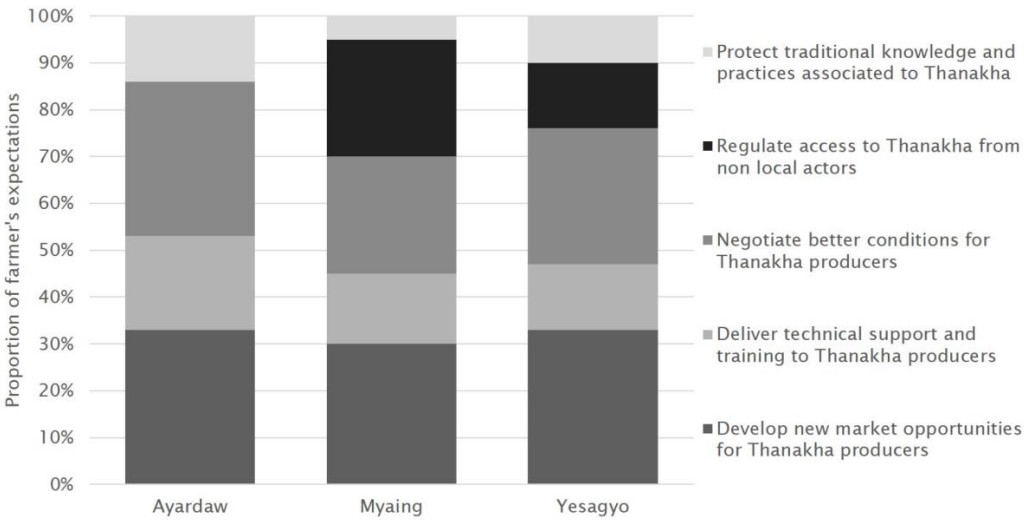

**Figure 9.** Farmers' expectations of the role of their farmers' associations.

### 3.2.3. Thanakha's Commercialization and Export Potential

This section reports on a small-scale study about Thanakha export potential. In Europe, as in northern America, Thanakha is currently only occasionally used in medicine or for skin treatments by very few companies. In order to increase the potential of Thanakha in these markets, actors like multinational cosmetic companies and fair-trade organisations

must find a commercial interest in this product. This can either be the Thanakha paste products as they are traditionally used in Myanmar, or new products based on Thanakha compounds. Marketing studies and commercial promotion may foster Thanakha's use among consumers outside Asia. Some key informants in Myanmar suggested that putting information like the origin and the traditional use on the final product's package could attract the interest of foreign consumers. According to the respondents, due to its skin-drying and cooling properties [29], Thanakha is highly appreciated by consumers in hot and humid countries. This is why they suggest orienting the potential export to tropical and sub-tropical countries. Thanakha products also have a potential market in high-purchasing-power countries' markets, such as in Europe and North America. However, this contradicts the principle of this product in Myanmar, where it is not a privilege for rich people. In terms of potential products, the respondents from Myanmar identified cosmetic cream and masks, medicine and sports massage emulsion as the most suitable Thanakha-based products for the export market in Europe and North America. The small-scale survey results indicate that Thanakha products have a relatively important potential to become a niche high-value trade product. Nevertheless, significant improvements are needed in the Thanakha value chain in Myanmar for it to be ready to tackle export markets for Europe and America. On the other hand, the main reasons expressed by the interviewed Swiss and French consumers to buy a potential Thanakha-based product were the fact that it is a natural product with many properties, which is linked to a long tradition, and has the potential to own a BioTrade label in the near future. The preferred potential Thanakha products mentioned were sun protection, skin imperfection creams (acne, pores) [31], and after-sport cream or gel to apply on the muscles. Other suggested preferred products were cosmetic face powder, after-sun soothing creams, and shampoo. All of the respondents indicated that it is important to find information about the origin, history and benefits of potential Thanakha products on the package. Among the few companies contacted during the study to assess their knowledge and interest in Thanakha, a company based in Geneva selling ethical and fair-trade handicrafts, food and cosmetics was the only one showing a great interest in Thanakha (Appendix A Box A3). Solid marketing studies to analyse the market potential of Thanakha products or compounds are still missing [31].

### 3.2.4. Thanakha's Genetic Resources and Associated Traditional Knowledge

Farmers value Thanakha, and in general the local biodiversity, because of the services and benefits they can obtain from it. The study results show that the respondents identified several types of benefits associated with Thanakha, and that the most valued benefits are associated with economic and income generation, as well as medicinal, skin care, cosmetic and beauty uses (Figure 10). This is in line with [31].

According to the respondents, these are followed by religious and cultural benefits, which highlights the strong link between Thanakha and Myanmar's culture and traditions.

An important portion of the respondents (82%) believed that the traditional knowledge associated with Thanakha is very important, and mentioned that few practices and TK associated with Thanakha are published or recorded. In this regard, there is a strong need to identify and recognize local communities' TK and their ownership rights.

A good example of a record associated with Thanakha's TK and practices is provided by the book 'Myanmar Thanaka' by U Thar Hla, 1974 (only available in Burmese). Traditional practices can be also attributed to farming activities based on traditional knowledge and local innovations, which are mainly transmitted orally, and are kept across generations. For example, a Thanakha farmer and owner of TK from Yesagyo township developed a vast knowledge of the consumer preferences related to Thanakha, and applied the traditional breeding of old varieties to develop a new variety from the Shin Ma Taung area to be appreciated on the market. The Farmers Associations declared that the TK and practices associated with Thanakha are being threatened. This is mostly explained by market-driven causes based on new trends in consumer preferences (particularly driven by young generations), which are more oriented to modern cosmetics. Thanakha farmers and traders

explicitly mentioned that it is extremely important that the government undertake the task of protecting both Thanakha's GR and its associated TK for future bioprospecting. The successful rationing of bioprospecting is considered to increase up by 400% thanks to TK [5,47].

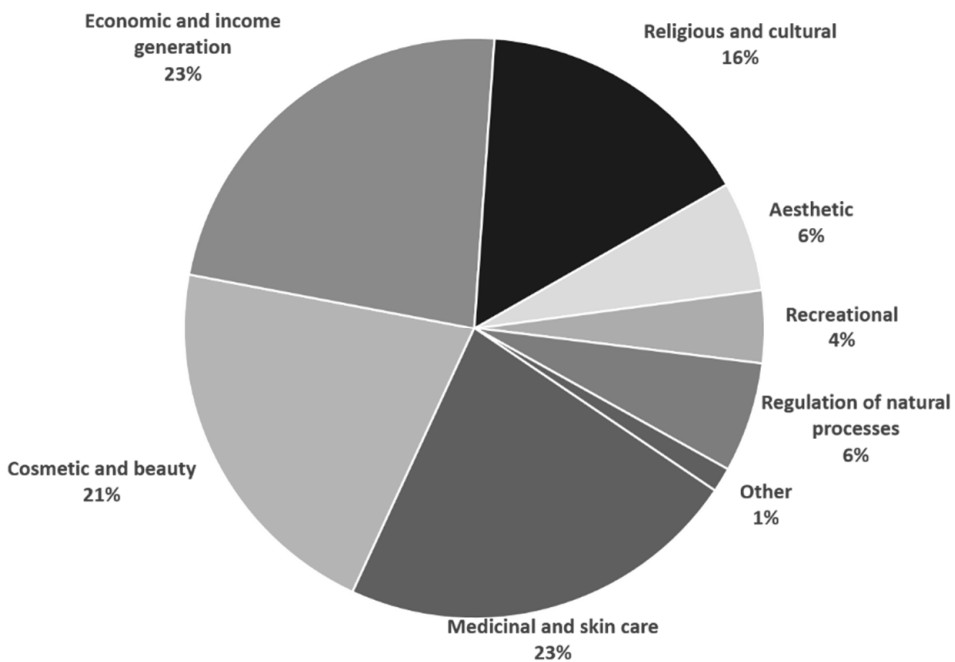

**Figure 10.** Benefits associated with Thanakha, as mentioned by farmers.

While the most valued benefits of Thanakha—i.e., in medicinal, skincare, and economic contexts—are very clear, its aesthetic and recreational benefits will only be explained by the biological diversity of the nature of Thanakha plants [48]. Thanakha is well known in Myanmar as a tropical treasure for its unique habitat of thriving in sandy soils in a hot and dry climate in which it is literally hard to grow any other commercial crops. The species, *Hesperathusa* (Naringi) *crenulata*, is described as a citrus variety by the College of Natural and Agricultural Sciences, University of California Riverside (UCR), and as "promising as an ornamental because of its beautiful, feathery, green foliage". [49]. For its oasis-like nature in the drylands, Thanakha has been viewed as having aesthetic and recreational values not only for human beings but also for local fauna, as reliable shelter.

### 3.2.5. Potential Equitable Benefit Sharing in the Thanakha Sector

There is a low level of awareness about the concept of benefit sharing among Thanakha stakeholders. Farmers, intermediaries and manufacturers lack knowledge about the existing rights provided by the Nagoya Protocol. One of the key factors that can measure the progress made in relation to equitable benefit sharing requirements is the level of access to information among producers sourcing raw material. Although 80% of the farmers declared that they receive some kind of market information, all of the farmers mentioned not being aware of whether Thanakha is being exported, or if the share of the value they receive is considered fair and equitable. Regarding price sale negotiation, the vast majority of the farmers (89%) mentioned having the last decision over the final price of sale, and over the decision to sell if no final price is agreed. Trust between farmers and buyers becomes an important factor during the negotiation of the sale price. This is mainly because buyers trust the farmers when they reveal the age of the trees that they are selling, which in turn is one of the main factors influencing the bark quality and the final price. Regarding the benefit sharing measures, our results suggest relevant opportunities for companies seeking to implement ethical sourcing practices. When the farmers were asked about possible benefits sharing opportunities regarding investments into local development, although

differing between townships, most of the priorities were focused on education infrastructure and electricity facilities, followed by health infrastructure, drinking water facilities and road connectivity (Figure 11). This is an important step in the implementation of ethical sourcing practices for the case of Thanakha in Myanmar in the framework of ABS regulations and practices.

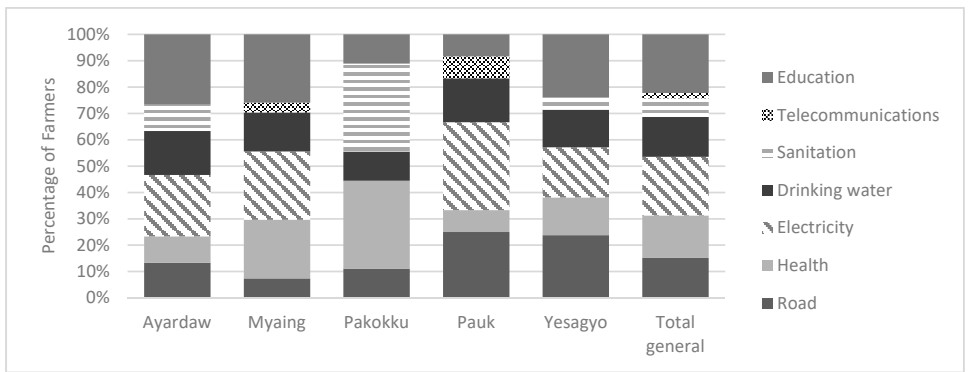

**Figure 11.** Farmers' demands for investments in rural infrastructure as benefit-sharing measures.

Regarding farmers' needs for the further development of Thanakha production activities, the three main issues that arose were the need for financial assistance (31%), manufacturing-supporting services (22%), and technical advisory (16%, with an additional 9% for extension). The needs for legal advice support, market information services and marketing services were poorly mentioned.

In the Thanakha production sector, the farmers argued that further market expansion would bring more possibilities for the implementation of protection measures for GR and traditional knowledge. In terms of collaboration between actors, private businesses currently performing R&D on Thanakha mentioned that they will share their results among the actors within the Thanakha value chain, which is a very important non-monetary benefit sharing mechanism.

## 4. Implications for the Implementation of ABS Measures for the Thanakha Sector

In 1995, Myanmar became party to the Convention on Biological Diversity (CBD), and since 2014 it has been party of the Nagoya Protocol [37]. However, a specific working regulatory framework associated with ABS requirements is still missing. A law and policy explicitly referring to access to GR and the fair and equitable sharing of the benefits arising from their use is still lacking. According to studies [12], a low institutional quality system—also measured by lacking IPR regulations—is negatively associated with a good probability of the implementation of the Nagoya Protocol system, no matter the time the country has been party to the Protocol.

As a first specific study objective, the current state and potential context of Access and Benefit Sharing in Myanmar has been analysed. The present study findings helped us to identify the constraints and opportunities related to the current ABS system in the country, using a SWOT Analysis (Table 3).

The national implementation of ABS measures from the Nagoya Protocol implies the adoption of legal, regulatory and policy actions, carrying out obligations through a standardized regulatory framework [12] regarding access to GR and TK, benefit-sharing obligations and compliance obligations, e.g., the application of monitoring systems. The implementation also implies the adoption of measures for the protection of TK, and designing institutional arrangements together with supporting measures like capacity-building initiatives.

As a second specific study objective, an analysis of the potential for the sustainable trade of biodiversity in Myanmar's ABS context was conducted, focusing on the Thanakha case.

In the Thanakha sector, the implications of ABS measures would become obligations for some stakeholders involved in the value chain. Besides the public institutions having the role of regulating the adoption of ABS requirements, other actors—such as companies—would have to review the relevance of their activities in accordance with the Nagoya Protocol. This would mean considering where ABS requirements are triggered in their bioprospecting activities [5]. An example would be the case of companies performing the extraction of marmesin from Thanakha as a bioactive compound capable of absorbing UV-A radiation.

**Table 3.** SWOT analysis of the current ABS system in Myanmar.

| STRENGTHS | WEAKNESSES |
|---|---|
| - Myanmar is in the Indo-Myanmar Biodiversity Hotspot and still owns a high level of biodiversity<br>- A few laws related to ABS have been just revised or are in the process of being reviewed<br>- Several ad-hoc permitting system to access GR in Myanmar are in place<br>- International initiatives are supporting the establishment of an ABS system | - Very low level of awareness and understanding of CBD and ABS system at all level<br>- Specific regulatory framework associated with ABS requirements is not in place<br>- No application of PIC or MAT in the R&D or commercial research agreements<br>- Lack of institutional communication<br>- Weak understanding on the roles of the ABS focal points |
| **OPPORTUNITIES** | **THREATS** |
| - Myanmar's high levels of biodiversity attracts scientific investigations to find natural substances for medicinal, cosmetic, pharmaceutical, and other uses<br>- International interventions contribute to fostering collaborations supporting the ABS implementation<br>- Some very motivated experts are willing to support the development of the ABS system<br>- Some ethnic communities (i.e., Karen) have strong customary rule about community forest sustainable management | - Natural resources management (and biodiversity) is one major reason for conflict in the peace and stability process<br>- Difficulty to involve a diversity of ethnic and other stakeholder groups (EAOs) in the discussions<br>- Loss of institutional memory and related capacity due to high turnover of staff in the Governmental institutions<br>- Lack of mutual trust between communities and authorities<br>- Lack of capacity in negotiating partnership agreements with foreign institutions |

If there are no relevant legislative or regulatory requirements, the implications would be associated with the implementation of the best practices on benefit sharing. In this case, the access to biodiversity and associated traditional knowledge for R&D should be subject to PIC and MAT, including a fair sharing of the benefits [44]. Under this scenario, other requirements linked to fair and equitable benefit sharing could be implemented, such as companies ensuring that the information used in sourcing negotiations is complete and accessible to the parties involved, as well as companies contributing to local sustainable development in the sourcing areas, as defined by producers and their local communities. For the targeting of the first measure, companies should work on improving their information transfer mechanisms for farmers, considering which type of information is relevant for them to promote production activities and so increase their income. For the second measure, companies should consider farmer's demand for rural infrastructure, especially the ones associated with increasing production capacities; hence, a territorial approach is of vital importance.

When approaching the recognition of TK associated with Thanakha, the following steps are necessary for the implementation of ABS measures.

(i)     Collect and consolidate available information regarding Thanakha TK
(ii)    Document verbal traditional knowledge through guidelines devel-oped together with the local community and recognized by the government
(iii)   Develop benefit sharing agreements with the identified and recognized TK owners.

As TK associated with Thanakha is owned by the whole country, identifying specific ownership would be challenging. In this case, traceability mechanisms that identify the origin from which the Thanakha was sourced could help to direct the benefit-sharing practices coming from the use of traditional knowledge.

Another implication of ABS implementation for actors involved in R&D activities with Thanakha would be associated with reaching an equal and fair distribution of shared value by identifying, in a participatory process, the best way to calculate the percentage of the final value of Thanakha cosmetics to be shared with local actors. However, challenges and opportunities for implementing ABS related to Thanakha exist (Table 4).

Finally, there are other potential tools that could protect local communities' rights to the specific biological resources of Thanakha, such as the Geographical Indications of origin (GI) and the BioTrade approach.

The initiative of developing a GI for Ayardaw's Thanakha dates back to 2014. At that time, it was agreed that Myanmar should generate intellectual property laws considering GI requirements.

For example, the well-renowned Shin Ma Taung Thanakha could also implement this tool. Farmers growing the Shin Ma Taung Thanakha variety agree that Thanakha grown in a 20-mile radius of the Shin Ma Taung mountain shares similar soil and climate conditions, and similar farming and cultural practices according to Thanakha traditional literature.

There is potential to create two Thanakha Geographical Indications of origin (GIs) in the Dry Zone. Farmers from Ayardaw Township are already aware of the benefits they could obtain from a GI label. They have local capacities to go through the process, as they are organized in associations [39]. Farmers from Myaing and Yesagyo could generate a GI for Thanakha originating from the renowned area of Shin Ma Taung. Both associations from each township could promote, in collaboration, the creation of the GI with the support of the Ministry of Commerce. The farmers mentioned that they already have some characteristics that could identify the profile of the GI product.

In the case of Thanakha in Myanmar, according to the study results, BioTrade can support the implementation of ABS requirements mostly by raising awareness of benefit sharing, promoting engagement, and providing guidance on the application of best practices. Even though there is no ABS framework in place, the case of Thanakha in Myanmar brings an interesting opportunity to strengthen the interface between BioTrade and ABS along a specific value chain, promoting alternative measures and benefit-sharing good practices in the same line as ABS requirements. The present study indicates that the Thanakha value chain may be sustainably promoted by implementing a BioTrade approach. This can be explained by several challenges and implications that exists for the implementation of an ABS system along the Thanakha value chain in Myanmar. For example, there is the challenge of protecting Thanakha TK through a national ABS regulatory framework, as Thanakha TK is already widely expanded among different countries, and foreign companies involved in R&D are already patenting Thanakha properties, which could be addressed by implementing BioTrade principles and ethical sourcing practices in the companies operating along the value chain [36].

**Table 4.** Challenges and opportunities for the implementation of ABS related to Thanakha.

| Opportunities | Challenges |
|---|---|
| • Strong social capital: at local level Thanakha Associations. play an important role to represent farmers in future negotiations of benefit sharing agreements. | • Awareness: Thanakha farmers are not aware about their rights to benefit from the use of biological resources and associated TK. |
| • Recognition of TK: Thanakha TK is part of Myanmar TKDL initiative. | • Land issue: Most of the farmers have upland category land title, which is destined only for growing annual agricultural crops. Growing only Thanakha is not allowed under this land tenure category. |
| • Thanakha traditional varieties: more than 17 varieties identified and kept at the Thanakha Museum. | • Negotiation skills: Thanakha farmers lack access to relevant information and negotiating skills to handle benefit sharing agreements. |
| • Breeders' rights: associated rights to some of these varieties could be protected through IPR under Traditional Variety Protection Laws, the Seed Law or other systems. | • Lack of traceability: Thanakha value chain is mostly based on informal transactions lacking traceability on the origin of sourcing and ethical sourcing practices. |

The promotion of local development through sourcing activities is another good practice regarding benefit sharing. In Myanmar, companies currently sourcing Thanakha material directly or indirectly are not compromised by the development of the local communities that the Thanakha producers inhabit. Although one national Thanakha-based cosmetics manufacturing company was mentioned to have sourcing preferences for farmers growing Thanakha naturally, promoting the protection of local biodiversity, there is no direct relationship between companies and farmers with respect to the direct delivery of support for sustainable local development. An indirect way in which national manufacturing companies are supporting local development is through the National Thanakha Federation, in which the inclusion of the Farmers' and Traders' Thanakha Associations strengthens local social capital and gives the possibility of dialogue about local development needs. Regarding local development needs, this study builds a baseline from which further negotiations can be developed, mainly in terms of rural infrastructure requirements and farmer's needs for the further production development of Thanakha. Briefly, while electricity, road connectivity and education are the major basic infrastructure demands from farmers, financial assistance, manufacturing supportive services, and technical advice are their main requests for the further development of Thanakha production activities. When planning and designing infrastructure investments in the Thanakha productive area in central Myanmar, it is important to consider a territorial approach based on the different demands between townships. As [50] mentioned, investment in basic rural infrastructure is one of the determinants for rural territorial development.

## 5. Conclusions

The present study investigated the current and planned regulations, systems and practices managing Access and Benefit Sharing in Myanmar.

The first specific study objective's findings on the current state and potential context of Access and Benefit Sharing in Myanmar identified a weak legal and institutional framework currently regulating and managing ABS implementation in Myanmar. Though Myanmar

became party to the Nagoya Protocol in 2014—as the first ASEAN country to do so—a specific regulatory framework associated with ABS requirements is not in place yet. As stated in [12]'s study, issues beside the timespan—such as the systems to protect natural environments and the quality of institutions—have an impact on the implementation of the Nagoya Protocol framework. However, none of the laws and policies directly and explicitly refer to access to GR and its associated TK, or the fair and equitable sharing of the benefits arising from their utilization. The Nagoya Protocol asks the signatory countries to develop national frameworks containing ABS requirements, and to regulate the country's GR and associated traditional knowledge [14,51]. However, in Myanmar, the implementation of a regulatory and institutional framework is still in the early stages of planning. The study shows that one of the main challenges the country is facing for the implementation of an ABS regulatory and institutional framework is the low level of awareness and understanding about ABS at the different stakeholder levels. This issue is also mentioned by several international initiatives targeting ABS implementation challenges around the world, e.g., the ABS Capacity Development Initiative, which argues that the relevant stakeholders are not aware of the existence of ABS, or are not informed about how an ABS system could be implemented at the national level.

A number of international projects have been supporting the country in the development an ABS framework, and some opportunities for the inclusion of Nagoya Protocol provisions inside national laws and policies are currently passing through a revision process.

Nevertheless, the level of awareness about ABS requirements and formal information about contracts and agreements is very low. This is in line with Smith et al.'s (2018) [44] findings about the struggles reported by organisations in complying with the Nagoya Protocol's requirements, and with receiving guidance from the National Focal Points and Competent National Authorities. Biocultural diversity composed of genetic resources and the associated traditional knowledge could be threatened in countries without ABS legislation in place [52]. Myanmar lacks a unique definition of TK which is recognized by all stakeholders. TK is normally transferred orally from one generation to another without being documented. Currently, no formal national mechanism is in place to document TK. It is extremely important to include traditional knowledge owners in the ecosystem governance dialogue in order to reach the goal of the conservation and sustainable use of biodiversity, long-term human wellbeing, and sustainable development [53,54].

The findings show that the creation of an ABS soft law may be a solution, while a full enforcement of the laws needs the involvement of the local communities' and ethnic minorities' trust in the Union Government. Local people and ethnic minorities' rights regarding GR and TK need to be protected. In order to support the Union Government, more work is required to shed light on the needs, i.e., conducting a full ABS gap analysis which can build on the work already performed by the previous projects and activities, as well as the recommendations on whether an administrative, policy or legislative measure is to be pursued. Based on this, Myanmar can decide which measure to pursue.

The findings related to the second specific study objective on the analysis of the potential for the sustainable trade of biodiversity in Myanmar's ABS context, focusing on the Thanakha case, shed light on the Thanakha-based cosmetics value chain in Myanmar. Thanakha is the bark of a tropical plant found in the Indian subcontinent and Southeast Asia ([28] *Hesperethusa crenulata* (syn. *Limonia acidissima*)), and has been traditionally used for the last 2000 years by Myanmar's people as a skincare and cosmetic product, and it has skin-caring and -protecting properties [29].

Thanakha's additional value is generated by processing the raw material into powder, and by manufacturing natural cosmetics containing Thanakha-based ingredients. R&D on GR is currently taking place regarding the manufacturing process of Thanakha-based cosmetics, in which foreign research institutions—sometimes in collaboration with local Thanakha factories—are currently performing the efficacy analysis of specific Thanakha properties, and are working on the extraction of specific biochemical compounds with promising market potential. According to [55], ABS obligations are triggered when R&D

over GR is performed, including research on the biochemical composition of biological resources which are also classified as derivatives [52], such as the active compound that protects against UV-A and can be used as an alternative to the currently available UV-B sunblock products.

Regarding the level of access to information, the findings show that Thanakha farmers do not have access to useful information for improvement into a more transparent and trustful negotiating process with their sourcing partners. This lack of access to relevant information could in turn increase the information asymmetry that often exists within value chains, and so could affect the power of local communities and producers to negotiate the benefits sharing on equal terms. Information symmetry helps farmers and local communities to access fair negotiations within value chain activities [56].

There is vast TK associated with Thanakha. Thanakha farmers own a crucial role in the preservation of traditional varieties and farming practices, which emphasizes the need to include them within the negotiations of local resources and TK management.

According to the study results, farmers place high importance on the TK associated with Thanakha, which is explained by the strong links between Thanakha and Myanmar's culture. The findings reveal that besides the widely renowned medicinal and cosmetic benefits of Thanakha, traditional knowledge can be attributed to the customary farming and breeding practices of old traditional Thanakha plant varieties. The study revealed several challenges to be tackled for the effective implementation of benefit-sharing practices, i.e., solving the current land tenure issues, raising awareness about benefit rights at the local level, improving farmers' access to information for fair negotiations, and improving the traceability mechanisms along the value chain. Regarding the challenging implementation of the ABS regulatory framework for Thanakha, the study provides insights into alternative benefit-sharing measures, such as protecting the rights of traditional varieties' breeders, the development of GI for Thanakha, or BioTrade. Thanakha products have a relatively important potential of becoming a high-value trade product to be commercialized on international markets. According to the small-scale survey findings, European consumers seem to be ready to try out products based on Thanakha. Potential thanakha-based creams and sun protection are the products which seem to generate the highest interest.

Understanding Myanmar's context regarding the specific regulatory framework around ABS implementation is of crucial importance for the safeguarding of the country's GR and TK, as well as the interest of the local communities, the custodians of these resources and the associated TK, like Thanakha, for which R&D bioprospecting activities for commercial purposes are bound to increase. This challenge calls for further and larger-scale studies on Thanakha as well as other GR and linked TK to limit the threats to their conservation and sustainable use.

**Author Contributions:** A.G. coordinated the study and led the preparation of the paper; S.M.A., facilitated all of the field work contacts and operations. S.M.A., J.T.U. and A.G. conducted the survey and field work on ABS. J.T.U. carried out the survey with the Thanakha farmers. T.D. conducted the small-scale survey on Thanakha's commercialization and export potential. A.G. and J.T.U. analysed the data. All of the authors contributed to writing the paper. All authors have read and agreed to the published version of the manuscript.

**Funding:** This study was funded—through a research consultancy mandate (No of project/mandate/country 1243.05.1.0)—by the Regional BioTrade Project South-East Asia, financed by the Swiss State Secretariat for Economic Affairs (SECO), and implemented by HELVETAS Swiss Intercooperation in Myanmar, Vietnam and Laos PDR.

**Informed Consent Statement:** Informed consent was obtained from all subjects involved in the study.

**Data Availability Statement:** The data presented in this study are available on request from the corresponding author. The data are not publicly available due to granting privacy of respondents.

**Acknowledgments:** Warm thanks go to all of the BioTrade Myanmar Team—Thet Thet Mar, Zaw Min Oo, and Khin Khin Soe—and to Andrew Wilson, former Manager of the Regional Biotrade Project Southeast Asia. Thanks also go to Rudolf Lüthi, Helvetas, Co-Head Water Food and Climate,

Helvetas Swiss Intercooperation, for providing a fundamental contribution to the study through technical and scientific direction. Sincere thanks are extended to all of the Thanakha farmers, and all of the interviewed people who took their time to share their precious knowledge and expert views.

**Conflicts of Interest:** The authors declare no conflict of interest.

## Appendix A

**Box A1.** Laws and policies regulating biodiversity conservation, and policies adopted to protect the environment in Myanmar.

| LAWS | Year |
|---|---|
| Environmental Conservation Law and subsidiary legislation | 2012 |
| Environmental Conservation Law | 2012 |
| Seed Law | 2011 |
| Myanmar Constitution | 2008 |
| Conservation of Water Resources and Rivers Law | 2006 |
| Traditional Medicine Law | 1996 |
| Protection of Wildlife and Conservation of Natural Areas Law | 1994 |
| Marine Fisheries Law | 1993 |
| Forest Law | 1992 |
| Freshwater Fisheries Law | 1991 |
| POLICY | |
| National Sustainable Development Strategy | 2009 |
| National Forestry Sector Master Plan | 2000 |
| Myanmar Agenda 21 | 1997 |
| Forest Policy | 1995 |
| National Environmental Policy | 1994 |

**Box A2.** The existing permit system to access GR in Myanmar.

1.  MoALI Agricultural Research Department (ARD): the MoALI ARD applies the Standard Material Transfer Agreement (SMTA) under the International Treaty on Plant Genetic Resources for Food and Agriculture (ITPGRFA) for the transfer of GRs from the seed bank to be shared with local and international organizations for research and propagation purposes (pers. comm MoALI-ARD Seed Bank). The collector does not share any benefits than those indicated in the SMTA. This point was identified as a constraint as stated in UNEP-China Trust Fund Project Final Report, as the level of benefits that can be achieved according to the SMTA under the ITPGRFA are limited [40]. Furthermore, to strengthen public awareness and participation in conserving plant genetic resources, the National Information Sharing Mechanism on the Implementation of Global Plan of Action for the Conservation and Sustainable Utilization of Plant Genetic Resources for Food and Agriculture [57], has been developed. The MoALI is also starting to work on plant variety protection (PVP) system for traditional varieties (pers. comm MoALI-ARD Seed Bank). A good example mentioned was the case of the Dutch company Rijkzwaan, which wanted to collect good genes of *Kasseti* cucumber variety (small melon) to be introduced in the melon varieties produced in Europe. Rijkzwaan is in the process to prepare an MoU with MoALI ARD to collect this variety. However, since the *kasseti* is a weed and it is not cultivated, it is impossible to sign an MoU to transfer this species, which is also not in the list ITPGRFA list. In this case, an ABS system would be appropriate. So, there were exchanges between the MoALI-ARD and the MoNREC-ECD. Yet, the MoNREC-ECD has not started to work on SMTA, and the Dutch company refuses to collect any species without the SMTA.

**Box A2.** *Cont.*

2.  MoNREC -Forest Research Institute, Forest Department (FRI): According to pers. comm. with the MoNREC Forest Research Institute, the FRI deals mostly with individuals and the system of permission that they give to access to forest GR from the World Heritage Sites is through institutional arrangements on a case by case basis. They expressed the need to do more systematic inventory of Myanmar's forest resources and they have existing arrangements on how this can be done. They conduct research on forest genetic resources in 68 districts, sustainable use of genetic resources, significant landscape conservation. They collect ethnobotany information on TK (on the use of GR and wildlife). They partnered with Japanese research institutes for an inventory of flora of Myanmar. The new Local community forest allows the allocation of land for forest community. Forest communities need to manage the land in a sustainable way. After a few years, communities can collect some of their forest GR and sell them within their township. Selling outside their township area, would imply the payment of a percentage on their sold produce to the local government.

3.  MoH, Department of Traditional Medicine (DTM): DTM works with about 3000 national companies involved in production and trading of TM (Fame Pharmaceuticals, Yokepyo Pharmaceutical company, Lwin Win Myint Aung & U Tha Yin Industries, Great Wall and Mawriya companies). Recent small collaboration with foreign institutes, like Miyazaki University and Toyama University of Japan have started to conduct joint research and capacity building. Through this MoU, the Japanese universities can collect medicinal plants and breed them in their labs. The DTM organizes the botanists who could accompany the Japanese researchers to Thanindary and Kahin State, looking for specific wild medicinal plants, through the contact with community's leaders, TM practitioners, and TM local authorities at state/province/township level. The collection of the medicinal plants is done for academic purpose. The Japanese research institute, on the other hand, grants training (Study tours for a number of Myanmar officers from DTM) in their universities on good manufacture practice of TM, and quality control, as well it informs on research findings, and offers co-authorship of research papers. Another Japanese institute, the Nakayama University, signed a collaboration agreement in 2017 with DTM for chemical analysis of natural compounds. The DTM does not apply any PIC, or MAT, or MTA. According to pers. comm. from DTM, they do not own much knowledge about ABS at DTM, but the department was invited to present the overview of the Myanmar ABS system in a conference in Thailand held in June 2018.

4.  MoE—Biotechnology Research Department (BRD): According to current staff at BRD, it was reported a limited knowledge and involvement in ABS related issues. BRD work involves research on plant and animal GRs in Myanmar. They stipulate MoU, with the International Foundation for Research grant, which provides the material for research and training and use of labs in Manitoba in return. The previous director who was involved in the UNEP-China trust fund project and the UNDP GEF V project reported that through many international collaborations, research and student exchange programmes, BRD is paying great care of transferring bio-samples and their bioactive components while conducting biotechnological research. By implementing some annexes of NP-ABS measures in MoU/contract of BRD, the BDR partner university and BRD itself can monitor the right to ABS of utilizing TK and GRs among parties, and applying articles n. 6, 8, 9, 10, 12, 17 and 23 of NP-ABS in enhancement of current MoU in BRD (pers. comm. BRD Former Director). Collaboration agreement between were signed between the BRD and the Yunnan Academy of Agricultural Sciences of China for conducting R&D activities on Myanmar's GR, in particular on rice.



**Box A3.** ABS implication for the promising export market of the Thanakha sector.

In relation to Thanakha as a high-export potential BioTrade product, the results from the Value Chain Analysis identified 5 main phases: production, trading, processing, manufacturing, and retailing. Value addition is generated mostly during processing and manufacturing activities, being in the last one where R&D for commercial purpose is currently taking place and ABS obligations could be triggered. The actors involved are: Forest Department dealing with permits, NGOs making commercial links, farmers' associations, local processors, input suppliers (packaging) from China, Universities (national and international) doing research, Unilever in Thailand and L'Oréal China, Department of Science and Technology that collaborates with R&D, Ministry of Commerce connecting with foreign R&D institutions and supporting associations, government agencies with mandate over natural resources, BRD from Ministry of Education, Thanakha manufacturers, IP Department, Department of Pharmaceutical from Ministry of Industry, MOALI, foreign investors, international organizations (ACB, UNDP, UNEP, GEF, etc.). Country where the export is destined are Thailand, India, Malaysia, Indonesia, Philippines, Vietnam, Australia, Germany, US, Singapore (processing company). Thanakha if often illegally trade at the boarders with neighbouring countries. The export format can be finished products, powder (foreign companies extract and export in liquid or solid formats).

In the Thanakha sector, the stakeholders mentioned that, although there is no collaboration platform or dialogue mechanism on Thanakha research issues, there are currently some collaborative initiatives going on. For example, the Thanakha Association of Ayardaw Township participated in one of the national ABS consultation workshops organized by the UNDP-GEF Global ABS were aware of the relevance of implementing ABS measures. Some Thanakha national companies doing R&D in collaboration with national universities, and international universities (Taiwan) and foreign (Chinese and Singapore) companies (and laboratories in Thailand, mainly on extraction efficacy, chemical properties and environmental product safety.

An interesting remark considering the impact that ABS implementation could have is associated with R&D initiatives is a Thanakha manufacturing company which is researching the efficacy of Thanakha properties and the extraction of specific compounds for commercial purpose. Using different techniques, researchers from South Korea could isolate the active compound *marmesin* from Thanakha bark, acting against UV-A, possible used as an alternative to UV-B protecting products. These discoveries create an enormous potential for the development of R&D in the Thanakha sector, hence, it is important having these activities aligned with the available regulating mechanisms regarding the use of biological resources in Myanmar.

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
