# Peer review of "Access and Benefit Sharing and the Sustainable Trade of Biodiversity in Myanmar: The Case of Thanakha"

_sustainability, doi:10.3390/su132212372_

Round 1
Reviewer 1 Report
In conclusion section (lines 898-902) the authors report: “The present study investigated on the current and planned regulations, systems and practices managing ABS in Myanmar. The study also analysed the potential of implementing ABS from the Nagoya Protocol or other measures for promoting sustainable development of Thanakha Value Chain as a high-export potential BioTrade product from Myanmar”.
- The present study is very interesting but the manuscript suffers in regards the quality of communication. Some improvements are needed in all sections. Thus, some selected comments are next given per section that can help the authors to conduct, as most as possible, improvements.
INTRODUCTION:
The authors state (lines 146-148): “The study aims to investigate on the current and/or planned regulations, systems and practices that will regulate and manage ABS in Myanmar. In addition, an investigation with particular focus on one of the potential BioTrade products in Myanmar (Thanakha) was conducted. The study is not an ABS Gap analysis.”
- Because the purpose of the study is too general and vague, the authors should clearly define it by adding some research objectives that enable them to properly align the results, discussion, implications and conclusions with the purpose of the study.
- They should eliminate some elements and information that nothing add to the manuscript as well as some repetition that there exists.
- They should also insert a short literature review to better highlight the research gap.
MATERIALS AND METHODS:
- I suggest authors to think to change the figure 1 to a table.
- Because a “combination of secondary and primary collected data and qualitative and quantitative tools were used to conduct the study” the authors should clearly define why they use the present combination of secondary and primary data as well as the qualitative and quantitative data.
- In lines 153 – 158 the authors state: “The literature review was based on review of both books and papers, as well grey literature, i.e. project reports from international and local research, non-governmental and governmental organizations and other specific ad-hoc documents provided by scientists and interviewed stakeholders during field work, and database and information repository like the Land, Agribusiness and Forestry Forum of Myanmar document repository which offers policy documents as well as a vast amount of published and unpublished grey literature on natural and human resources in Myanmar”. I suggest authors to: 1) clearly define all kinds of the references they use discussing also their usefulness and reliability, 2) clearly define the main reasons they have collected qualitative and quantitative data, 3) highlight what exactly each category of data/information provides to authors.
- In regards the data collection, they should sufficiently explain why only 35 interviews have been conducted (it is too small sample) as well as why they have chosen only 20 consumers interviews,
- The authors should present the research design because it could facilitate them to properly present the results as well as the readers to better understand the work.
- They should also eliminate the repetition there exist, e.g. table 1.
RESULTS
- The authors should clearly present the processed data as well as the literature on which each subsection is based. It is suggested they to follow the research objectives step by step, taking into consideration the descriptions presented in "Materials and methods" section, that concern the literature review and data collection (qualitative and quantitative).
- They could partially reorganize the section in order the results to be in alignment with the objectives of the study.
- Some information that nothing add to the manuscript is included in the results section. Certain elements and information can be eliminated or/and can be moved in an Appendix.
- The authors should think to properly shorten some of the subsections, such as “Institutional framework for biodiversity and ABS matters in Myanmar” and “Regulatory framework and current developments related to ABS” as well as “Recent and current ABS supporting activities in Myanmar” and “ABS institutional and policy framework”.
- Because the samples (of the quantitative data) are too small, the percentages the authors note are not good indicators to interpret and communicate the results and adequately discuss them. For example, in lines 721-722, the authors note: “A few farmers mentioned the need of legal advice support (6%), market information services (5%) and marketing services (5%). Six percent of the interviewed expressed other needs.” The authors should take into account that the percentage 6% reflects 2 farmers as well as the percentage 5%!
- There is no a systematic presentation of the data that are collected. Authors should present the most important results in one or more tables as well as highlight the data they analyze in each subsection.
- The results should be presented following the individual research objectives. Alternatively, the authors could follow the appropriate research hypotheses they could set.
CONCLUSION
- A SWOT analysis has been included in conclusions. It could be better to move it into the results or discussion or implications.
REFERENCES AND TITLE
- The authors should follow the journal guidelines to properly cite and report the references.
- They could also think to partially change the title of the article in order to clearly reflect the content of the manuscript.
Author Response
Dear Reviewer, Many thanks for your thoughful and very useful feedback that helped us to improve the manuscript. Following your explanatory suggestions on each point, we heavily modified some parts with the aim to improve the overall paper as per your advice. See our answers after each of your suggestions (in red). Language editing has also been applied: .
INTRODUCTION:
1- The authors state (lines 146-148): “The study aims to investigate on the current and/or planned regulations, systems and practices that will regulate and manage ABS in Myanmar. In addition, an investigation with particular focus on one of the potential BioTrade products in Myanmar (Thanakha) was conducted. The study is not an ABS Gap analysis.”- Because the purpose of the study is too general and vague, the authors should clearly define it by adding some research objectives that enable them to properly align the results, discussion, implications and conclusions with the purpose of the study.
Two specific objectives have been included at the end of the introduction chapter. These objectives should now clarify our research objectives. The methodology chapter (methods) and the results chapter are also aligned with these objectives.
2- They should eliminate some elements and information that nothing add to the manuscript as well as some repetition that exists.
The manuscript has been shortened to eliminate some less relevant elements to the objective, and the repetition. All the changes can be seen in track changes.
3 - They should also insert a short literature review to better highlight the research gap.
We largely expanded our literature review adding references from peer reviewed (the new references are highlighted in yellow the list of references) and specific literature review has been added (for Myanmar biodiversity, ABS, Thanakha)
MATERIALS AND METHODS:
4- I suggest authors to think to change the figure 1 to a table.
We left it as a Figure, because we think that it is more descriptive in the present state of a figure.
5- Because a “combination of secondary and primary collected data and qualitative and quantitative tools were used to conduct the study” the authors should clearly define why they use the present combination of secondary and primary data as well as the qualitative and quantitative data.
We added extensive additional explanatory text clarifying these points in the Materials and Methods chapter.
6- In lines 153 – 158 the authors state: “The literature review was based on review of both books and papers, as well grey literature, i.e. project reports from international and local research, non-governmental and governmental organizations and other specific ad-hoc documents provided by scientists and interviewed stakeholders during field work, and database and information repository like the Land, Agribusiness and Forestry Forum of Myanmar document repository which offers policy documents as well as a vast amount of published and unpublished grey literature on natural and human resources in Myanmar”.
suggest authors to: 1) clearly define all kinds of the references they use discussing also their usefulness and reliability, 2) clearly define the main reasons they have collected qualitative and quantitative data, 3) highlight what exactly each category of data/information provides to authors.
- We greatly expanded the literature with new references, mainly peer reviewed papers (see additional references in red colour).and added how we used them.
- The reasons for collecting both qualitative and quantitative data is explained in the part of Materials and Methods (see also our feedback to your point 5 above)
- this information was also added in the section Material and Methods
7- In regards the data collection, they should sufficiently explain why only 35 interviews have been conducted (it is too small sample) as well as why they have chosen only 20 consumers interviews,
We have better explained it now in the text (Mateirals and Methods), and hereafter:
- The Thanakha production area in Myanmar is quite restricted to the dry area and to 5 townships. Our study included Thanakha actors from all of the five townships and took advantage of the existing farmer’s organizations which were present to facilitate the data collection process. The limitation of the area and the specification of the study on THanakha as potential BioTrade product in compliance with ABS in Myanmar and the qualitative nature of the study justifies the small sample of the respondents.
- As reported in the manuscript, the potential consumers’ survey was done on a very little scale and on a case study basis to get an idea on consumers’ interest and its potential commercialization in Europe. The case study is not a marketing study and it is not representative, but it gives a preliminary indication of the possible consumers’ attitude towards a product very far from their culture. Please see the paragraph we added at the end of the Materials and Methods.
8 - The authors should present the research design because it could facilitate them to properly present the results as well as the readers to better understand the work.
We added two specific objectives (see also the above point 1), which are now included in the introduction and explained in the Materials and Methods, and the Result chapter is also aligned with them as well. We think that in this way (thanks to your previous suggestion), the paper gained clarity to guide the readers now.
9 - They should also eliminate the repetition there exist, e.g. table 1.
We deleted Table 1 as suggested. We eliminate repetitions in the text.
RESULTS
10 - The authors should clearly present the processed data as well as the literature on which each subsection is based. It is suggested they to follow the research objectives step by step, taking into consideration the descriptions presented in "Materials and methods" section, that concern the literature review and data collection (qualitative and quantitative).
The two sections of the results (3.1 and 3.2) now start a sentence clarifying both the specific objective of the study and the type of data collected
11- They could partially reorganize the section in order the results to be in alignment with the objectives of the study.
Each section of the results (3.1 and 3.2) now starts with the reference to the specific objective it refers to. The specific objectives are now explained in the last paragraph of the Introduction, as clarified in the above point 1
12- Some information that nothing add to the manuscript is included in the results section. Certain elements and information can be eliminated or/and can be moved in an Appendix.
The manuscript has been shortened, by eliminating main parts in the Result sections, and some parts have been moved to the Appendix, as suggested
13 - The authors should think to properly shorten some of the subsections, such as “Institutional framework for biodiversity and ABS matters in Myanmar” and “Regulatory framework and current developments related to ABS” as well as “Recent and current ABS supporting activities in Myanmar” and “ABS institutional and policy framework”.
The above mentioned sections have been shortened and heavily amended, by eliminating non essential parts to increase readability, please see track changes,
14 - Because the samples (of the quantitative data) are too small, the percentages the authors note are not good indicators to interpret and communicate the results and adequately discuss them. For example, in lines 721-722, the authors note: “A few farmers mentioned the need of legal advice support (6%), market information services (5%) and marketing services (5%). Six percent of the interviewed expressed other needs.” The authors should take into account that the percentage 6%
we improved the presentation of the results, avoiding the general assumptions based on non-representative data.
15 - There is no a systematic presentation of the data that are collected. Authors should present the most important results in one or more tables as well as highlight the data they analyze in each subsection.
Now the results follow better the specific objectives reported at the end of the Introduction chapter. We revised a table, added tables and a figure (Figure 10) and Appendices to report the results, We changed the title of some Figures reporting the results for a better explanation
16- The results should be presented following the individual research objectives. Alternatively, the authors could follow the appropriate research hypotheses they could set.
Now the results follow better the specific objectives reported at the end of the Introduction chapter.
CONCLUSION
17- A SWOT analysis has been included in conclusions. It could be better to move it into the results or discussion or implications.
We moved the SWOT in the implication part, which is not Table 3 and we edited some of the SWOT points
REFERENCES AND TITLE
18 - The authors should follow the journal guidelines to properly cite and report the references.
The citations and the list of references follow the journal guidelines now
19 - They could also think to partially change the title of the article in order to clearly reflect the content of the manuscript.
We proposed a new title: Access and Benefit Sharing and the sustainable trade of biodiversity in Myanmar: the case of Thanakha
Reviewer 2 Report
Sustainability_1428029_Review Note
Access and Benefit Sharing in Myanmar and the Thanakha case
Overall comments
The study investigates on the current and planned regulations and practices managing Access and Benefit Sharing (ABS) in Myanmar focusing on one of the potential BioTrade products: Thanakha. This is one of the good contributions in this area of research. However, the manuscript in the current form is poor in terms of its scientific rigour and needs thorough revisions. Consistencies, paragraph sequencing, thematic focus, clarity in sentences and referencing are the key aspects to be revised.
Reference is not in the format of MDPI
Abstract
natural-based products or nature-based products?
- Introduction
- Bioversity of Myanmar or Biodiversity in Myanmar ? (74)
- Too long and detailed introduction but mostly focused on generic context. It would be good if authors could shortened the generic context and add some specific information of Thanaka and its background/context (also the subject of the study)
- Objective is clear but I suggest not to present objective in the separate sub-heading. Can be included as a part of Introduction chapter.
- Materials and Methods
UNCLEAR--------The survey sites were selected by the Helvetas Myanmar BioTrade team, based on level farmers are currently collaborate in an associative manner at township level (Undurraga 2018).
What was the methods and framework of content analysis ?- would be good to detail further.
- Results and discussion
Too long results and discussions section, specially first two subsections.
Sub section 3.2.1 Lines 627-630 very confusing, unclear sentences. E.g What authors mean by …The implementation of Thanaka ???- does not make sense to me.
Authors highlighted several benefits from Thanaka as their findings of the study. However, these are too generic. For example, sub section 3.2.5 authors have also mentioned aesthetic benefits and recreational benefits from Thanaka, that is not clear. Would be good if authors could first define the type of benefits and then rank or rate the benefits based on the respondent’s view (table or figure would be a benefit to the readers).
Figure 10 and its narration on the priority of the local development seems not much relevant considering the focus of the paper.
Implications
Implications section is relatively OK. The issues highlighted in this section should be linked with the results. Currently, this aspect is poor. Moreover, this section needs more references and evidences to support the claims/points of the authors.
Conclusions
This section is too long and NOT within the framework of writing scientific conclusion. The table in this section looks odd.
Author Response
Dear Reviewer, Many thanks for your thoughful, precise and useful feedback. Following your direction on each point, we heavily modified and edited the whole manuscript, also expanding our literature adding a number of citations (which we lighlighted in yellow in the list of references). Language editing has also been applied. See our answers (in red) following each of your suggestions: .
1 - Reference is not in the format of MDPI
The citations and the list of references follow the journal guidelines now
Abstract
2- natural-based products or nature-based products?
Natural products ‘ corrected in the abstract and first paragraph
Introduction
3-Bioversity of Myanmar or Biodiversity in Myanmar ? (74)
Corrected into ‘Biodiversity in Myanmar’, it was a typo.
4-Too long and detailed introduction but mostly focused on generic context. It would be good if authors could shortened the generic context and add some specific information of Thanaka and its background/context (also the subject of the study)
We shortened and reaorganised he introduction on general context and we added specific information and references in the section Biodiversity of Myanmar.
5-Objective is clear but I suggest not to present objective in the separate sub-heading. Can be included as a part of Introduction chapter.
Objectives have been further clarified and embedded in the Introduction without a separate sub-heading
Materials and Methods
6- UNCLEAR--------The survey sites were selected by the Helvetas Myanmar BioTrade team, based on level farmers are currently collaborate in an associative manner at township level (Undurraga 2018).
Now it has been clarified in the manuscript
7 What was the methods and framework of content analysis ?- would be good to detail further.
We added a clarification on the method, including the software used at the end of the chapter
Results and discussion
8 - Too long results and discussions section, specially first two subsections.
The chapter has been greatly shortened, by eliminating main parts in the Result sections in particular in the first 3.1 sub-section, but also reorganising the presentation of the findings in 3.2..
9- Sub section 3.2.1 Lines 627-630 very confusing, unclear sentences. E.g What authors mean by …The implementation of Thanaka ???- does not make sense to me.
Would that be section 3.2.3? we paraphrased that sentence (which was previously 627-630), modified the ‘implementation of Thanakha’ sentence and improved the whole 3.2.3 section by heavily editing it.
10 - Authors highlighted several benefits from Thanaka as their findings of the study. However, these are too generic. For example, sub section 3.2.5 authors have also mentioned aesthetic benefits and recreational benefits from Thanaka, that is not clear. Would be good if authors could first define the type of benefits and then rank or rate the benefits based on the respondent’s view (table or figure would be a benefit to the readers).
The 3.2.5 section has been heavily modified to address the reviewer’s comment above. A figure has been added to show the type of benefits of Thanakha and the rate of the benefits based on the respondent’s view (table or figure would be a benefit to the readers).
Aesthetic benefits and recreational benefits from Thanakha have been clarified, see the last paragraph of 3.2.5
11- Figure 10 and its narration on the priority of the local development seems not much relevant considering the focus of the paper.
We decided to keep Figure 10 (which is Figure 11 in the revised version) as the figure shows the different benefit sharing opportunities mentioned by the respondents. This is an important step in the ABS implementation, or at least in the implementation of ethical sourcing practices. More than helping to identify ABS regulations or practices, the figure contributes towards exploring the potential of ABS in our case of study: Thanakha in Myanmar. We changed the title of the Figure for clarity and added an explanatory sentence. See the changes in 3.2.6.
Implications
12 - Implications section is relatively OK. The issues highlighted in this section should be linked with the results. Currently, this aspect is poor. Moreover, this section needs more references and evidences to support the claims/points of the authors.
We added more references to support our points in the discussion (included in the Result and Discussion chapter). We focused the implications directly on the potential implementation of ABS on Thanakha (see change of the title of the chapter) and followed the specific objectives of the study, now highlighted at the end of the Introduction chapter, and also added further references.
Conclusions:
13 - This section is too long and NOT within the framework of writing scientific conclusion. The table in this section looks odd.
We greatly modified the conclusions. We moved the SWOT table in the implication part, which is now Table 3 and we edited some of the SWOT points
Reviewer 3 Report
The subject of the article is interesting, and it is linked to the objectives of the journal, however, there are many issues that have to be reconsidered.
For better visibility on databases, the authors are asked not to repeat among keywords the words/concepts included in the title of the article.
Line 20. please use thanakha with capital letter
The paper is too long and difficult to follow. I recommend to shorter it and eliminating non-essential information.
The Conclusion part is far too long, it should be more concise and it should offer a clear response to the research statements/aims.
Author Response
Dear Reviewer, Many thanks for your thoughful and useful feedback. Following your advice, we greatly modified, amended and edited. Literature review has been enriched by a number of citations (now highlithed in yellow). Please see track changes in the text and the new references highlighted in the reference list.. Language editing has also been applied. See our answers after each of your suggestions (in red): .
1- For better visibility on databases, the authors are asked not to repeat among keywords the 33- words/concepts included in the title of the article.
We added a keyword, but we also left the previous keywords and we understand from the Sustainability guidelines, that some words from the title can be repeated in the keywords
2 - Line 20. please use thanakha with capital letter
Done
3 - The paper is too long and difficult to follow. I recommend to shorter it and eliminating non-essential information.
The manuscript has been greatly shortened and heavily amended, by eliminating non essential parts in all the chapters, and to increase readability, please see track changes,
4-The Conclusion part is far too long, it should be more concise and it should offer a clear response to the research statements/aims.
We greatly modified the conclusions. We moved the SWOT table in the implications part, which is now Table 3 and we edited some of the SWOT points
Round 2
Reviewer 1 Report
Dear Editors
The authors elaborated on the manuscript and made substantial improvements, following our most important comments and suggestions. They have also sufficiently responded to our comments and suggestions, point by point. Thus, we suggest the manuscript to be accepted for publication.
Author Response
Thanks for your kind feedback. Again, many thanks for your detailed recommendations on how to improve our manuscript. We did some minor editing, and we improved the quality of some figures (original size figures have been submitted).
Reviewer 2 Report
Thanks to the Authors. Comments are nicely addressed and the manuscript is improved.
Author Response
Thanks for your kind feedback and for your advice and comprehensive suggestions on how to improve our manuscript. We did some minor editing, and we improved the quality of some figures (original size figures have been submitted).
Reviewer 3 Report
The authors did comprehensive work of improving the manuscript.
Still, the keyword repeate many the words/concepts from the title (e.g. Access and Benefit Sharing, Biodiversity, Myanmar, Thanakha), which is not advisable.
Author Response
Thanks for your kind feedback. Many thanks for your suggestions on how to improve our manuscript. We did some minor editing, and we improved the quality of some figures (original size figures have been submitted).
Following your good advice of changing keywords, avoiding the words reported in the title, we proposed new set of keywords as follows:
ABS Institutional framework, BioTrade, Burmese Thanaka, Traditional Knowledge, Genetic Resources, Natural ingredients, Value chain, Nagoya Protocol, Hesperethusa crenulate/Limonia acidissima
In the keyword we spelt Thanakha in a different way as in the text (Thanaka) as both names are correct, and we thought this could increase the visibility.